# THE LLM SURGEON

**Tycho F.A. van der Ouderaa**[1*]**, Markus Nagel**[2]**, Mart van Baalen**[2]**,**
**Yuki M. Asano**[3]**, Tijmen Blankevoort**[2]
[1]Imperial College London , [2]Qualcomm AI Research[†], [3]QUVA Lab, University of Amsterdam

## ABSTRACT

State-of-the-art language models are becoming increasingly large in an effort to achieve the highest performance on large corpora of available textual data. However, the sheer size of the Transformer architectures makes it difficult to deploy models within computational, environmental or device-specific constraints. We explore data-driven compression of existing pretrained models as an alternative to training smaller models from scratch. To do so, we scale Kronecker-factored curvature approximations of the target loss landscape to large language models. In doing so, we can compute both the dynamic allocation of structures that can be removed as well as updates of remaining weights that account for the removal. We provide a general framework for unstructured, semi-structured and structured pruning and improve upon weight updates to capture more correlations between weights, while remaining computationally efficient. Experimentally, our method can prune rows and columns from a range of OPT models and Llamav2-7B by 20%-30%, with a negligible loss in performance, and achieve state-of-the-art results in unstructured and semi-structured pruning of large language models. Code is available at: https://github.com/Qualcomm-AI-research/llm-surgeon.

## 1 INTRODUCTION

Recent advancements in language modeling (Vaswani et al., 2017) allow fitting large language models (LLMs) with millions or even billions of parameters (such as OPT (Zhang et al., 2022) and Llama 2 (Touvron et al., 2023)) on big text corpora achieving high performance. Unfortunately, the size of these LLMs often makes it hard to deploy them within practical constraints. Cloud-based deployment can get very expensive for larger models, and efficient devices such as phones are frequently limited in the memory size to host a model.

A body of literature extending back to the late 1980s, e.g., Optimal Brain Damage (OBD, LeCun et al. (1989)) and Optimal Brain Surgeon (OBS, Hassibi & Stork (1992)), phrases pruning as a constraint optimization problem to reduce a model's footprint and runtime requirements. The Hessian required for this approach grows with the square of the number of parameters, and can only be computed in practice for unrealistically small networks. To overcome this issue, Eigendamage (Wang et al., 2019) introduces a Kronecker factorization of a blockwise-diagonal approximation of the Hessian. Recent works, like Optimal Brain Compression (Frantar & Alistarh, 2022), SparseGPT (Frantar & Alistarh, 2023), demonstrate practical post-training pruning of LLMs, but only consider a loss curvature of a pruned layer's squared output reconstruction error, ignoring gradients that relate local removal costs to the target loss. As a result, their approximation to the target loss landscape is inaccurate, leading to a significant performance degradation for pruned LLMs. Further, these methods do not readily extend to structured pruning.

---

[*]Work done while doing an internship at Qualcomm AI Research
[†]Qualcomm AI Research is an initiative of Qualcomm Technologies, Inc.

Structured compression (rows and columns)  Unstructured compression (matrix elements)

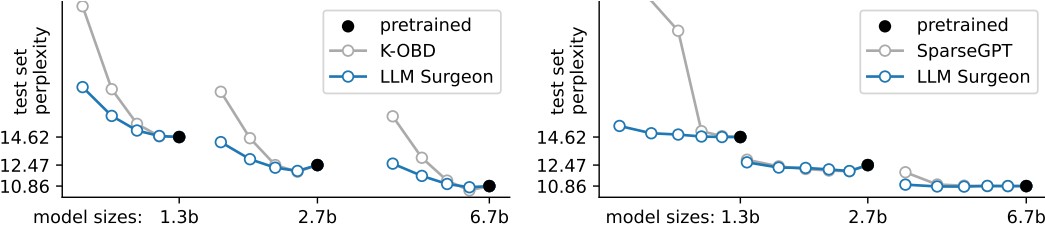

Figure 1: LLM Surgeon allows interpolation of model size between existing pretrained models.

This work introduces LLM Surgeon, a general framework for unstructured, semi-structured and structured pruning of LLMs. At paper submission, we deemed this the first method to successfully perform structured pruning of LLMs. Concurrent work by Ashkboos et al. (2024) also considers structured pruning of LLMs but ignores gradient information, resulting in lower final performance. The superior performance of LLM Surgeon is achieved by scaling up the block-diagonal Kronecker-factorized approximations to the empirical Fisher from Eigendamage to LLMs. We further expand upon the work by deriving OBS-like weight pruning costs and updates for structured pruning of multiple rows and columns, and provide a general framework that also incorporates semi-structured and unstructured pruning. Instead of treating individual weight updates independently, we strive to consider as many correlations between weights as practically possible and derive joint weight updates for pruning multiple weights (or multiple sets of structured weights) at once. Unlike prior work in LLM pruning, LLM Surgeon prunes in multiple shots, updating weights and curvature estimates between shots. We use global thresholding for unstructured, semi-structured and structured, i.e., instead of pruning layers by a fixed amount, more sensitive layers are pruned less than those that are more robust. Lastly, we propose to mitigate possible first-order gradients not being zero by using optional low-rank first-order updates between shots. A key advantage of LLM Surgeon is that it allows trading off additional compute during compression for better accuracy by increasing the number of correlations and/or shots. Our method gives the first practically usable results for structured pruning of LLMs – they can be pruned by up to 30% with minor performance degradation. Furthermore, we achieve state-of-the-art results in unstructured and semi-structured LLM pruning.

## 2 BACKGROUND AND RELATED WORK

Neural network pruning aims to remove parameters from a model while minimizing negative impact on final performance. More formally, we denote the $P$ model parameters as vector $\boldsymbol{\theta}^* = \text{vec}(\boldsymbol{W}_1^*, \boldsymbol{W}_2^*, \dots \boldsymbol{W}_L^*) \in \mathbb{R}^P$, by flattening the $L$ weight matrices of attention and fully-connected blocks, with already fitted $\boldsymbol{\theta}^* \approx \arg\min_{\boldsymbol{\theta}} \mathcal{L}(\boldsymbol{\theta})$ to data $\mathcal{D}$ to minimise a negative likelihood loss $\mathcal{L}(\boldsymbol{\theta}) = -\log p(\boldsymbol{\theta}|\mathcal{D})$. To compress the model, we are looking for a pruned vector $\hat{\boldsymbol{\theta}}$:

$$\hat{\boldsymbol{\theta}} = \arg\min_{\boldsymbol{\theta}} \mathcal{L}(\boldsymbol{\theta}) \text{ s.t. pruning constraints based on } \boldsymbol{\theta}^* \tag{1}$$

where chosen constraints determine the structure of compressed weights $\hat{\boldsymbol{\theta}}$. In **unstructured pruning**, a fraction of total weight elements is set to zero. In **semi-structured pruning** of M:N we have that M weights of every N consecutive weights are zero (Zhou et al., 2021; Hubara et al., 2021). And in **structured pruning** (Louizos et al., 2017), entire rows and columns are set to zero. Structured pruning leads to the most immediate gains in memory and computing, as it directly reduces the dimensions of matrices that need to be represented explicitly but is regarded as the most difficult to compress. Maintaining high performance is often easier in the other schemes but requires specialised arithmetic exploiting the sparsity structure to benefit at deployment. We consider all pruning types above, with a focus on structured pruning for LLMs.

Typically, eq. (1) can not be solved directly, as the space of possible pruning configurations exceeds what can be evaluated in practice. To illustrate, a search over all possible unstructured pruning masks of a 125 million parameter LLM would require $2^P = 2^{125m} \approx 10^{37628749}$ evaluations. The idea, therefore, is to find $\hat{\boldsymbol{\theta}}$ using a surrogate of the loss landscape $q$ that is easier to work with:

$$\mathcal{L}(\boldsymbol{\theta}) = -\log p(\mathcal{D} \mid \boldsymbol{\theta}) \approx -\log q(\boldsymbol{\theta}) \tag{2}$$

If one chooses a particular Gaussian form for our surrogate $q$, then solutions for unstructured, semi-structured, and structured pruning constraints can be derived in closed-form (appendix A).

### 2.1 TAYLOR EXPANSION

How do we obtain a good surrogate of the loss $q$? One of the easiest approaches is to locally expand the log loss through a second-order Taylor expansion around the pretrained weights $\boldsymbol{\theta}^*$, yielding:

$$-\log q(\boldsymbol{\theta}) \approx -\log p(\mathcal{D}|\boldsymbol{\theta}^*) - (\boldsymbol{\theta} - \boldsymbol{\theta}^*)^T \nabla \mathcal{L}(\boldsymbol{\theta}^*) - \frac{1}{2}(\boldsymbol{\theta} - \boldsymbol{\theta}^*)^T \boldsymbol{H}_{\boldsymbol{\theta}^*}(\boldsymbol{\theta} - \boldsymbol{\theta}^*) \tag{3}$$

where $[\nabla \mathcal{L}(\boldsymbol{\theta}^*)]_i = \frac{\partial}{\partial \boldsymbol{\theta}_i} \mathcal{L}(\boldsymbol{\theta}_i^*)$ denotes the Jacobian and $[\boldsymbol{H}_{\boldsymbol{\theta}}]_{ij} = \frac{\partial^2}{\partial \boldsymbol{\theta}_i \boldsymbol{\theta}_j} \mathcal{L}(\boldsymbol{\theta}_{ij})$ denotes the Hessian. The first-order term vanishes $[\nabla \mathcal{L}(\boldsymbol{\theta}^*)]_i = \boldsymbol{0}$ at the optimum. Note that in practice the first order

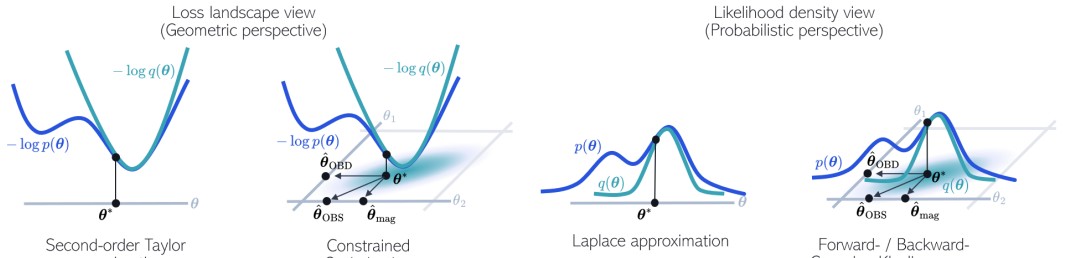

Figure 2: Pruning as equality constrained optimization of quadratic approximation of the loss landscape (left), or equivalently, maximising the likelihood under a Laplace approximation (right).

term may not vanish. While we follow this assumption initially, we consider interleaved first-order corrections to mitigate the issue in section 3.6. The quadratic expansion of eq. (3) forms the basis of the optimal brain damage (LeCun et al., 1989) and optimal brain surgeon (Hassibi & Stork, 1992) pruning methods. Note that from a probabilistic perspective, a quadratic approximation of the log likelihood implies a Gaussian approximation of the likelihood, as also observed by (Wang et al., 2019) and illustrated in fig. 2. This is well-known (Bishop & Nasrabadi, 2006), (MacKay, 2003) as the Laplace approximation $q(\boldsymbol{\theta}) = \mathcal{N}(\boldsymbol{\theta} \mid \boldsymbol{\theta}^* + \nabla\mathcal{L}(\boldsymbol{\theta}^*), \boldsymbol{H}_{\boldsymbol{\theta}^*}^{-1})$, with pretrained weights are the mean and the local inverse Hessian is the covariance matrix capturing *correlations between weights*.

## 2.2 BLOCK FISHER INFORMATION MATRIX

For a network trained with negative log-likelihood loss, the Hessian is identical to the Fisher matrix:

$$\boldsymbol{H}_{\boldsymbol{\theta}} = \boldsymbol{F}_{\boldsymbol{\theta}} = \sum\nolimits_{n=1}^{N} \mathbb{E}_{y \sim p_{\boldsymbol{\theta}}(y|x_n)} \left[ \nabla_{\boldsymbol{\theta}} \log p_{\boldsymbol{\theta}}(y|x_n) \nabla_{\boldsymbol{\theta}} \log p_{\boldsymbol{\theta}}(y|x_n)^T \right] \tag{4}$$

which has the benefit of always being positive semi-definite, with the inverse thus forming a proper covariance matrix for $q$, and can be approximated with Monte Carlo samples of $p_{\boldsymbol{\theta}}(y|x_n)$. For most LLMs, this would be treating the softmax output of the network as categorical distribution $p_{\boldsymbol{\theta}}(y|x_n)$, and sampling from that. In practice, we use the 'empirical Fisher' replacing the expectation over $y$ with target data $y_n$ (Kunstner et al., 2019). The full (empirical) Fisher $\boldsymbol{F}_{\boldsymbol{\theta}} \in \mathbb{R}^{P \times P}$ scales quadratically in the number of parameters $P$. To overcome this, the Fisher is often written in terms of layer-wise blocks $\boldsymbol{F}_{lk} = \sum_{n=1}^{N} \mathbb{E}\left[ \text{vec}(\nabla_{\boldsymbol{W}_l} \log p_{\boldsymbol{\theta}}(y|x_n))\text{vec}(\nabla_{\boldsymbol{W}_k} \log p_{\boldsymbol{\theta}}(y|x_n))^T \right]$, and approximated by only treating layers independently (Martens & Grosse, 2015; Botev et al., 2017):

$$\boldsymbol{F}_{\boldsymbol{\theta}} = \text{diag}(\boldsymbol{F}_{11}, \boldsymbol{F}_{22}, \ldots, \boldsymbol{F}_{LL}), \qquad \boldsymbol{F}_l = \sum\nolimits_{n=1}^{N} \mathbb{E}\Big[ \underbrace{(\boldsymbol{g}_{l,n}\boldsymbol{g}_{l,n}^T) \otimes (\boldsymbol{a}_{l,n}\boldsymbol{a}_{l,n}^T)}_{RC \times RC} \Big] \tag{5}$$

where $\otimes$ denotes the Kronecker product and $\text{vec}(\cdot)$ the matrix vectorisation operation. Because we disregard cross-layer interactions we write $\boldsymbol{F}_l$ instead of $\boldsymbol{F}_{ll}$ for Fisher blocks associated with the weight matrix $\boldsymbol{W}_l \in \mathbb{R}^{R \times C}$ producing outputs $\boldsymbol{y}_{l,n} = \boldsymbol{W}_l \boldsymbol{a}_{l,n} \in \mathbb{R}^R$ from inputs $\boldsymbol{a}_{l,n} \in \mathbb{R}^C$, for each layer $l$ and datapoint $n$. Consequently, we can compute Fisher blocks from input activations $\boldsymbol{a}_{l,n} \in \mathbb{R}^C$ of forward-passed data $x_n$ and output gradients $\boldsymbol{g}_{l,n} = \nabla_{\boldsymbol{y}_{l,n}}\mathcal{L} \in \mathbb{R}^R$ from backpropagation.

## 2.3 PRUNING AS CONSTRAINED OPTIMIZATION

Optimal brain surgery relies on removing and adapting weights such that the loss is least negatively affected, thus it behooves us to write the problem as a constrained optimization problem. From the Gaussian approximation discussed in section 2.1 obtained by quadratically expanding the log likelihood loss $-\log p \approx \frac{1}{2}\boldsymbol{\theta}^T\boldsymbol{F}\boldsymbol{\theta}$, the optimal update $\Delta\boldsymbol{\theta} = \hat{\boldsymbol{\theta}} - \boldsymbol{\theta}$ (and thus also $\hat{\boldsymbol{\theta}} = \boldsymbol{\theta} + \Delta\boldsymbol{\theta}$) becomes the following equality constrained quadratic optimization problem (Hassibi & Stork, 1992):

$$\underset{\Delta\boldsymbol{\theta}}{\arg\min} \ \frac{1}{2}\Delta\boldsymbol{\theta}^T\boldsymbol{F}\Delta\boldsymbol{\theta} \tag{6}$$
$$\text{s.t. } \boldsymbol{e}_k^T\Delta\boldsymbol{\theta} + \boldsymbol{e}_k^T\boldsymbol{\theta} = 0, \forall k \in \mathcal{K}$$

where $\boldsymbol{F}$ is positive semi-definite and $\mathcal{K}$ is the set of $K$ indices that are pruned (i.e., set to zero).

**Algorithm 1** LLM Surgeon (*structured*)

---

**Input:** initial weights $\boldsymbol{\theta}^0$, target size $\alpha$, and data $\mathcal{D}$
   **For** shot $t$ in $[1, 2, \ldots, T]$
        **Compute:** approximate curvature $\boldsymbol{G}$, $\boldsymbol{A}$ from data $\mathcal{D}$                 $\triangleright$ section 3.1
        **Compute:** costs per row/column $\mathcal{L}_r, \mathcal{L}_c$ from $\boldsymbol{G}$, $\boldsymbol{A}$         $\triangleright$ section 3.2
        **Compute:** threshold $\tau$ using $\mathcal{L}_r$ and $\mathcal{L}_c$ given target size $\alpha_t$     $\triangleright$ section 3.3
        **Select:** rows and columns to remove $\boldsymbol{E}_R$, $\boldsymbol{E}_C$ based on $\tau$        $\triangleright$ section 3.3
        **Compute:** weight update $\Delta\boldsymbol{\theta}^{t-1}$ based on $\boldsymbol{E}_R, \boldsymbol{E}_C$ and $\boldsymbol{G}, \boldsymbol{A}$     $\triangleright$ section 3.4
        **Update:** remaining weights $\boldsymbol{\theta}^t \leftarrow \boldsymbol{\theta}^{t-1} + \Delta\boldsymbol{\theta}^{t-1}$            $\triangleright$ section 3.5
        **Optionally:** $\boldsymbol{\theta}^t \leftarrow$ low-rank update$(\boldsymbol{\theta}^t)$                $\triangleright$ section 3.6
   **Output:** compressed weights $\hat{\boldsymbol{\theta}} = \boldsymbol{\theta}^T$

---

**General solution** We denote $\boldsymbol{E}_K = \begin{bmatrix} \boldsymbol{e}_1 & \boldsymbol{e}_2 & \ldots & \boldsymbol{e}_K \end{bmatrix}^T \in [0,1]^{K \times P}$ as a matrix of which the row vectors are canonical basis vectors $\boldsymbol{e}_k \in \mathbb{R}^P$ that select the elements to be pruned. One of the most standard approaches to solve eq. (6) is using Langrange multipliers, which results in a general closed-form solution for the expected increase in loss $\mathcal{L}$ and optimal weight update $\Delta\boldsymbol{\theta}$:

$$\mathcal{L} = \frac{1}{2}(\boldsymbol{E}_K \boldsymbol{\theta}^*)^T \left(\boldsymbol{E}_K \boldsymbol{F}^{-1} \boldsymbol{E}_K^T\right)^{-1} \boldsymbol{E}_K \boldsymbol{\theta} \tag{7}$$

$$\Delta\boldsymbol{\theta} = -\boldsymbol{F}^{-1} \boldsymbol{E}_K^T \left(\boldsymbol{E}_K \boldsymbol{F}^{-1} \boldsymbol{E}_K^T\right)^{-1} \boldsymbol{E}_K \boldsymbol{\theta} \tag{8}$$

which we use to derive unstructured, semi-structured, structured for modern Fisher approximations (see appendices A.2 to A.4). The same general form of eqs. (7) and (8) appears in prior LLM pruning work Kurtic et al. (2022), but only for much simpler layer-wise pruning and no structured pruning.

## 3 LLM SURGEON

This section describes the components of our method, **LLM Surgeon**, summarised in algorithm 1.

### 3.1 ESTIMATING LOSS LANDSCAPE CURVATURE

Accurate pruning relies on approximating the local curvature accurately while overcoming the memory cost associated with storing the true curvature. Specifically, even with the block-wise approximation of eq. (5), $\boldsymbol{F} \in \mathbb{R}^{RC \times RC}$ requires summing $N$ large $RC \times RC$ matrices, too large to practically fit in memory. Instead, we adapt the KFAC approximation (Martens & Grosse, 2015) that assumes independence of activations and derivatives, approximating an expectation of Kronecker products as a Kronecker product of two expectations $\mathbb{E}[\boldsymbol{g}_{l,n}\boldsymbol{g}_{l,n}^T \otimes \boldsymbol{a}_{l,n}\boldsymbol{a}_{l,n}^T] \approx \mathbb{E}[\boldsymbol{g}_{l,n}\boldsymbol{g}_{l,n}^T] \otimes \mathbb{E}[\boldsymbol{a}_{l,n}\boldsymbol{a}_{l,n}^T]$, allowing layer-wise Fisher blocks to be approximated as $\boldsymbol{F}_l \approx \widetilde{\boldsymbol{F}}_l$, where

$$\widetilde{\boldsymbol{F}}_l = \boldsymbol{G}_l \otimes \boldsymbol{A}_l \quad \text{, with } \boldsymbol{G}_l = \frac{1}{\sqrt{N}} \sum_{n=1}^N \boldsymbol{g}_{l,n} \boldsymbol{g}_{l,n}^T \text{ and } \boldsymbol{A}_l = \frac{1}{\sqrt{N}} \sum_{n=1}^N \boldsymbol{a}_{l,n} \boldsymbol{a}_{l,n}^T \tag{9}$$

constructed from activations $\boldsymbol{a}_{l,n} \in \mathbb{R}^C$ from forward passes and gradients $\boldsymbol{g}_{l,n} \in \mathbb{R}^R$ from backward passes (Eschenhagen et al., 2024). The approximation originates from optimization literature, but has recently gained popularity for other problems that require curvature approximations (Immer et al., 2022; van der Ouderaa et al., 2023), including structured pruning in Wang et al. (2019).

An additional advantage of approximating Fisher blocks as Kronecker products is that the inverse becomes particularly easy to compute $\widetilde{\boldsymbol{F}}^{-1} = \boldsymbol{G}^{-1} \otimes \boldsymbol{A}^{-1}$, thus only requires inverting the factors. This fact allows us to never explicitly construct large $RC \times RC$ matrices in memory that make up $\widetilde{\boldsymbol{F}}$ and $\widetilde{\boldsymbol{F}}^{-1}$, but rather directly work with the much smaller matrices $\boldsymbol{G}$ and $\boldsymbol{A}$.

### 3.2 COMPUTING COSTS IN FINAL LOSS

The number of possible combinations in which weights can be removed grows (supra-)exponentially in parameter count, making it infeasible to estimate a separate cost $\mathcal{L}$ for each such removal. A common strategy, therefore, is to treat weights independently when computing removal costs $\mathcal{L}$. We also follow this strategy, but note that this does not necessarily imply that we have to make such same strong independence assumption for the weight updates $\Delta\boldsymbol{\theta}$ after selecting weights to be removed.

Unlike most prior work, we present correlated weight updates by taking into account off-diagonal elements of the Fisher approximation in section 3.4.

For semi-structured and unstructured we use independent costs for individual weight elements $k \in [1, RC]$, and for structured use independent costs for all rows $r \in [1, R]$ and columns $c \in [1, C]$. We find that we can derive the appropriate costs from the general cost formula eq. (7) by letting $\boldsymbol{E} = \boldsymbol{e}_k \in \mathbb{R}^{RC}$ where the single one-hot element at index $k$ of canonical basis vector $\boldsymbol{e}_k$ selects the weight to remove. For structured pruning, we similarly select rows $r$ and columns $c$, by setting $\boldsymbol{E} = \boldsymbol{e}_r^T \otimes \boldsymbol{I} \in \mathbb{R}^{C \times RC}$ or $\boldsymbol{E} = \boldsymbol{I} \otimes \boldsymbol{e}_c \in \mathbb{R}^{R \times RC}$ with $\boldsymbol{e}_r \in \mathbb{R}^R$, $\boldsymbol{e}_c \in \mathbb{R}^C$. Plugging into eq. (7), we find:

$$\mathcal{L}_k = \frac{1}{2} \frac{(\boldsymbol{\theta}_k)^2}{[\boldsymbol{G}^{-1} \otimes \boldsymbol{A}^{-1}]_{kk}}, \qquad \mathcal{L}_r = \frac{1}{2} \frac{\boldsymbol{\theta}_r^T \boldsymbol{A} \boldsymbol{\theta}_r}{[\boldsymbol{G}^{-1}]_{rr}}, \quad \mathcal{L}_c = \frac{1}{2} \frac{\boldsymbol{\theta}_c^T \boldsymbol{G} \boldsymbol{\theta}_c}{[\boldsymbol{A}^{-1}]_{cc}} \qquad (10)$$

Full derivations can be found in appendices A.2 and A.3. The costs for single elements $\mathcal{L}_k$ are equivalent to those found in optimal brain surgeon (Hassibi & Stork, 1992) and $\mathcal{L}_r$ and $\mathcal{L}_c$ closely resemble structured brain surgeon of (Wang et al., 2019), but in our case derived for matrix rows and columns (see appendix A.3). Given curvature estimates, costs for either removing all weights or all rows and columns can be computed in parallel. In addition, we derive costs for the more general sum of Kronecker factor approximation $\widetilde{\boldsymbol{F}} \approx \boldsymbol{G}_1 \otimes \boldsymbol{A}_1 + \boldsymbol{G}_2 \otimes \boldsymbol{A}_2$ in appendix I through an eigendecomposition.

### 3.3 DYNAMIC WEIGHT ALLOCATION WITH GLOBAL THRESHOLD

Unlike prior works that compress layer-by-layer (Frantar & Alistarh, 2023), we use a global threshold $\tau$ enabling a *dynamic allocation of sparsity levels across layers*, pruning most where it hurts the least. Our method can compress a model to a specifically chosen target size $\alpha$, defined as the fraction of weights that should remain, i.e. stay non-zero after compression. In all structured, semi-structured, and unstructured pruning (fig. 3), we select as many

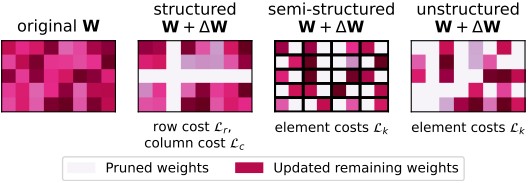

Figure 3: General framework for structured, semi-structured and unstructured compression.

weights for removal so that the target size $\alpha$ is reached that inflict the least possible costs $\mathcal{L}$, as computed according to section 3.2. For unstructured pruning, this is as simple as sorting the costs for all weights $\mathcal{L}_k$ in the network and setting a global threshold $\tau$ such that $\alpha$ fraction of weights fall within the threshold $\mathcal{L}_k \leq \tau$. For M:N semi-structured pruning, we sort the M costs of each N consecutive weights and select the M weights with lowest cost. In case of a multi shot schedule (see section 3.5) we also sum the M lowest costs in each block to find a cost per block, sort costs per block across the entire network, and similar to the unstructured case set a global threshold $\tau$ such that an $\alpha$ fraction of weights fall within threshold. Lastly for structured pruning, we perform a sorting appropriately weighted by the number of elements that make up a row or column and set the global threshold $\tau$ such that $\alpha$ fraction of all weights fall within the threshold. Then we remove all rows and columns that fall within the threshold $\mathcal{L}_r, \mathcal{L}_c \leq \tau$.

### 3.4 CORRELATED WEIGHT UPDATES

Like most other pruning methods, we prune multiple weights at once (Frantar & Alistarh, 2023; Wang et al., 2019). To arrive at pruning costs and weight updates for pruning multiple weights, it is common to compute costs and updates for individual weights (or sets of weights) independently and add them together to arrive at a joint pruning cost. In LLM Surgeon, we argue that it's better to consider weight updates jointly instead of independently. After selecting the set of weights for pruning, we can often afford to compute a single correlated weight update associated to the joint removal of multiple weights, instead of naively summing weight updates associated to individual removals. We derive such correlated weight updates below. Note that, for the expected cost computation, we do assume that the row, column or weight costs are independent, as the number of possible combinations of weights to prune grows too large to compute within reasonable time.

**Fast unstructured / semi-structured correlated weight updates** Mathematically, we represent pruned weights as $\boldsymbol{E}_K = [\boldsymbol{e}_1 \quad \boldsymbol{e}_2 \quad \ldots \quad \boldsymbol{e}_{R'}]^T \in \mathbb{R}^{K \times RS}$, where $\boldsymbol{e}_r \in \mathbb{R}^{R'}$ are one-hot canonical ba-

sis vectors selecting the weights for removal. As each element $k$ has a unique associated row $r$ and column $c$ index, we can consequently also use canonical basis vectors for these respective rows $\boldsymbol{E}_R \in \mathbb{R}^{K \times R}$ and columns $\boldsymbol{E}_C \in \mathbb{R}^{K \times C}$ (i.e., we have $[\boldsymbol{E}_R]_i \otimes [\boldsymbol{E}_C]_i = [\boldsymbol{E}_K]_i$ is satisfied for all $i$).

We derive unstructured weight updates in appendix A.2, by considering eigendecompositions $\boldsymbol{G} = \boldsymbol{K}_1 \boldsymbol{S}_1 \boldsymbol{K}_1^T$, $\boldsymbol{A} = \boldsymbol{K}_2 \boldsymbol{S}_2 \boldsymbol{K}_2$ of the Fisher approximation $\boldsymbol{F} \approx \boldsymbol{G} \otimes \boldsymbol{A}$, which from eq. (8) yields:

$$\Delta \boldsymbol{W} = \boldsymbol{G}^{-1} \Big( \boldsymbol{K}_1 \Big( \underbrace{\overline{\boldsymbol{K}_1^T} \overline{\boldsymbol{W}}^{-1} \overline{\boldsymbol{K}}_2 \oslash \boldsymbol{S}}_{K \times K} \Big)^{-1} \boldsymbol{K}_2 \Big) \boldsymbol{A}^{-1} \tag{11}$$

where $\oslash$ is element-wise division, and for brevity use bar notation $\overline{\boldsymbol{K}}_1 = \boldsymbol{E}_K \boldsymbol{K}_1$, $\overline{\boldsymbol{K}}_2 = \boldsymbol{E}_K \boldsymbol{K}_2$, $\overline{\boldsymbol{\theta}} = \boldsymbol{E}_K \boldsymbol{\theta}$, and $\boldsymbol{S} = \mathrm{diag}(\boldsymbol{S}_1)\mathrm{diag}(\boldsymbol{S}_2)^T \in \mathbb{R}^{R \times C}$, and $\mathrm{diag}(\cdot)$ vectorises matrix diagonals.

Programmatically, we always avoid explicitly representing large matrices $\widetilde{\boldsymbol{F}}$ and $\widetilde{\boldsymbol{F}}^{-1}$ in memory, but rather compute relevant quantities from their factors. Likewise, we never represent sparse matrices $\boldsymbol{E}_K$, $\boldsymbol{E}_R$ or $\boldsymbol{E}_C$ in memory, but instead work with a lists of indices of the one-hot elements directly. For example, we can cheaply construct $\overline{\boldsymbol{K}}_1 = \boldsymbol{E}_R \boldsymbol{K}_1 \in \mathbb{R}^{K \times R}$ and $\overline{\boldsymbol{K}}_2 = \boldsymbol{E}_C \boldsymbol{K}_2 \in \mathbb{R}^{K \times C}$, by copying row vectors, and the vector $\overline{\boldsymbol{\theta}} = \boldsymbol{E}_K \boldsymbol{\theta} = \boldsymbol{E}_R \boldsymbol{W} \boldsymbol{E}_C^T \in \mathbb{R}^K$ by indexing all pruned weights.

**Maximum number of correlated weights** The main computational bottleneck is the $K \times K$ matrix inverse in eq. (11). To control compression speed, we can split pruned weights into disjoint subsets $K = K_1 \cup K_2 \cup \ldots$, such that each subset $K_i$ does not exceed the set maximum number of correlated weights $K_i \leq m$, and sum associated independent updates. Using less correlation by setting a lower $m$ allows trading compression quality for speed.

**Fast structured correlated weight updates** Unlike the general case which requires inverting a $K \times K$ matrix for $K$ correlated weights, we find that weight updates with the Kronecker factored Fisher approximation $\tilde{\boldsymbol{F}} = \boldsymbol{G} \otimes \boldsymbol{A}$ only require inverting a $R' \times R'$ matrix when removing $R'$ rows or a $C' \times C'$ matrix when removing $C'$ columns. The updates are much cheaper than we would have expected based on the effective number of weights in those rows and columns, which would imply inverting $R'C \times R'C$ or $RC' \times RC'$ matrices. In practice, this leads to a significant speed-up for structured pruning and weight updates that take into account correlations between rows or columns. When removing $R'$ rows, $r_1, r_2, \ldots r_{R'}$, or the $C'$ columns $c_1, c_2, \ldots, c_{C'}$, with $1 < R' < R$ and $1 < C' < C$, we denote one-hot vectors selecting all rows and columns to be removed respectively as $\boldsymbol{E}_{R'} = \begin{bmatrix} \boldsymbol{e}_1 & \boldsymbol{e}_2 & \ldots & \boldsymbol{e}_{R'} \end{bmatrix}^T \in \mathbb{R}^{R' \times R}$ and $\boldsymbol{E}_{C'} = \begin{bmatrix} \boldsymbol{e}_1 & \boldsymbol{e}_2 & \ldots & \boldsymbol{e}_{C'} \end{bmatrix}^T \in \mathbb{R}^{C' \times C}$. We find weight updates associated to removing the $R'$ rows by setting $\boldsymbol{E}_K = \boldsymbol{E}_{R'} \otimes \boldsymbol{I}$ or $\boldsymbol{E}_K = \boldsymbol{I} \otimes \boldsymbol{E}_{C'}$:

$$\begin{aligned} \text{remove multiple } R' \text{ rows:} \qquad & \Delta \boldsymbol{W} = -\overline{\boldsymbol{W}} (\boldsymbol{E}_{C'} \boldsymbol{A}^{-1} \boldsymbol{E}_{C'}^T)^{-1} (\boldsymbol{A}^{-1} \boldsymbol{E}_{C'}^T) \\ \text{remove multiple } C' \text{ columns:} \qquad & \Delta \boldsymbol{W} = -\boldsymbol{G}^{-1} \boldsymbol{E}_{R'}^T (\boldsymbol{E}_{R'} \boldsymbol{G}^{-1} \boldsymbol{E}_{R'}^T)^{-1} \overline{\boldsymbol{W}} \end{aligned} \tag{12}$$

From here, it is clear that the special case of removing a single row $r$ or column $c$ under Kronecker approximation involves inverting a $1 \times 1$ matrix, and thus only requires scalar division:

$$\text{remove single row } r: \ \Delta \boldsymbol{\theta} = -\frac{\boldsymbol{G}^{-1} \boldsymbol{e}_r \otimes \boldsymbol{\theta}_r}{[\boldsymbol{G}^{-1}]_{rr}} \ , \text{ or single column } c: \ \Delta \boldsymbol{\theta} = -\frac{\boldsymbol{\theta}_c \otimes \boldsymbol{A}^{-1} \boldsymbol{e}_c}{[\boldsymbol{A}^{-1}]_{cc}} \tag{13}$$

in accordance to independent structured updates in Wang et al. (2019), for convolutional filters. We have thus extended existing structured weight updates to rows and columns, and derived update rules that also consider correlation between structured groups (in our case the rows and columns).

### 3.5 MULTI SHOT PRUNING SCHEDULE

To improve the performance-to-sparsity ratio, we propose pruning in multiple shots. We theoretically justify this multi-shot approach by noting that the surrogate loss landscape $q$ relies on a Taylor expansion (eq. (3)) that only holds locally and thus becomes unreliable for larger jumps $\Delta \boldsymbol{\theta}$ in parameter space. We mitigate this by pruning in multiple $T > 1$ shots, $t \in [1, 2, \ldots, T]$, each resulting in a smaller weight update $\Delta \boldsymbol{\theta}$ after which the curvature of the loss surface can be re-estimated. When pruning to target size $\alpha$, ie. removing $1 - \alpha$ of total weights, we choose a schedule $\alpha_t$ starting at $\alpha_0 = 1$ and ends with $\alpha_T = \alpha$, such that after $T$ shots, exactly $\alpha$ fraction of the total weight remain. Empirically, we find that a linear schedule for $\alpha_t$, as formulated in section 4, monotonically improves

Table 1: Structured compression of large language models on wikitext-2 data.

| Method | Target size | Test performance (PPL) | | | | |
| --- | --- | --- | --- | --- | --- | --- |
| | | OPT (125m) | OPT (1.3b) | OPT (2.7b) | OPT (6.7b) | Llama-v2 (7b) |
| Baseline | 100% | **27.65** | **14.62** | **12.47** | **10.86** | **5.12** |
| Magnitude | 90% | 767.2 | 894.4 | 1229 | 3464 | 36746 |
| $I \otimes I$ | 80% | 4685 | (1278) | 2788 | 16747 | 347960 |
| | 70% | 17970 | (3098) | 9255 | 17312 | 41373 |
| L-OBD | 90% | 33.3 | 20.76 | 17.69 | 27.20 | 14259 |
| diag($I \otimes A$) | 80% | 94.14 | 1392 | 3236 | 7570 | 15630 |
| multi shot | 70% | 545.6 | 2147 | 7233 | 7628 | 21386 |
| K-OBD | 90% | 27.97 | 14.68 | 11.96 | 10.53 | 5.48 |
| diag($G \otimes A$) | 80% | 29.89 | 15.63 | 12.47 | 11.28 | 9.14 |
| multi shot | 70% | 36.54 | 18.29 | 14.53 | 13.03 | 15.43 |
| | 60% | 47.54 | 24.65 | 18.09 | 16.21 | 28.03 |
| | 50% | 75.95 | 37.68 | 26.68 | 25.54 | 46.64 |
| LLM Surgeon (**ours**) | 90% | 28.29 | 14.73 | 12.00 | 10.82 | 5.43 |
| $G \otimes A$ | 80% | 29.37 | 15.27 | 12.37 | 11.22 | 7.29 |
| within row/col cor. $\Delta$ | 70% | 32.46 | 16.60 | 13.16 | 11.83 | 10.85 |
| | 60% | 39.82 | 19.40 | 14.79 | 12.94 | 16.67 |
| | 50% | 51.48 | 23.81 | 18.01 | 15.38 | 25.62 |
| LLM Surgeon (**ours**) | 90% | 28.01 | 14.70 | 12.02 | 10.77 | 5.25 |
| $G \otimes A$ | 80% | 28.73 | 15.12 | 12.27 | 11.02 | 6.18 |
| full cor. $\Delta$ | 70% | 31.82 | 16.24 | 12.92 | 11.64 | 7.83 |
| | 60% | 38.47 | 18.45 | 14.23 | 12.58 | 10.39 |
| | 50% | 49.78 | 22.95 | 17.15 | 14.90 | 15.38 |

pruning performance with more shots, and that higher sparsity levels typically require more shots (see appendix F.1). Multi-shot pruning allows one to spend (linearly in $T$) more computation to improve the final compression performance.

## 3.6 INTERLEAVED LOW-RANK FIRST-ORDER CORRECTIONS

We propose optional interleaved low-rank first-order corrections to further improve compression performance. So far, we assumed parameters are in a local optimum when finding a closed-form solution to the quadratic constraint problem. In practice, however, this assumption likely does not hold since (i) the neural network may not be optimised to the minimum, (ii) a different loss may be used for compression than used for training, or (iii) we prune in multiple shots (section 3.5) inevitably causing weights to diverge from the optimum. To mitigate this, we consider first-order corrections by interleaving pruning shots with low-rank adaptations of weights $W_l + UV$ (LoRA, by (Hu et al., 2021)), commonly used in LLM finetuning. We always absorb updates after each shot, so that the next loss estimate $q$ is closer to the optimum and underlying assumptions are likely to hold more closely. By absorbing LoRA updates between shots, the sum of low-rank updates can have a higher rank than individual updates. That is, we have rank($U^1 V^1 + U^2 V^2 + \ldots + U^T V^T$) $\geq$ rank($U^t V^t$) for the updates $U^t V^t$ at any shot $t$, with equality only arising if updates lie exactly in the same subspace which is unlikely to ever occur in practice. This insight could also be used during regular LoRA finetuning and may therefore be useful outside the context of model compression to allow more expressive low-rank model adaptation, at negligible cost.

## 4 RESULTS

We compare compression performance of LLM Surgeon on language modeling tasks on OPT (Zhang et al., 2022) and Llama-v2 (Touvron et al., 2023) model families, using data from wikitext-2 dataset (appendix B.2). For compression, we use 128 sequences with a sequence length of 2048 tokens from the training data set and evaluate test perplexity (PPL) on the standard test split. In our experiments, we use a linear sparsity schedule $\alpha_t = 1 - t(\frac{1-\alpha}{T})$ at each shot $s$ before reaching the final sparsity $\alpha$. We use 40 shots at $\alpha = 0.5$ sparsity and report intermediate compression rates, effectively using $T = 8$ shots for $\alpha = 0.9$, $T = 16$ for $\alpha = 0.8$, $T = 24$ for $\alpha = 0.7$, and $T = 32$ for $\alpha = 0.6$. We compare against magnitude pruning, L-OBD, SparseGPT and K-OBD baselines. The K-OBD and LLM Surgeon use the multi shot procedure of section 3.5 using $T = 40$ shots for structured pruning and $T = 5$ shots for semistructured and unstructured pruning. Further details are found in appendix B.

## 4.1 Structured Compression

Structured compression of rows and columns enables direct savings in memory and compute through a straight reduction of matrix dimensions in the model. For LLM surgeon, we consider in section 3.4 weight updates with different levels of correlations: limited to correlations within rows and columns, and correlations both within and between rows and columns. We further compare against magnitude pruning, which only uses weight magnitudes, L-OBD, which only uses activations, and K-OBD, which also uses Kronecker-factored curvature but assumes full independence and thus only prunes without updating remaining weights. We report results in table 1, and observe that more correlations results in better performance, with the largest improvements for the Llama-v2 model family.

While a 50% structured compression is not better than a smaller model of similar size, LLM Surgeon allows us to reduce model size by up to 30% with minimal loss, without training a smaller model from scratch fig. 1. In our structured compression experiments our proposed LLM Surgeon method outperforms all baselines and achieves the best performance for each compression target size.

## 4.2 Interleaved low-rank updates

Additionally, we assess compression performance in conjunction with the proposed first-order corrections using the interleaved low-rank adaptation described in section 3.6. We find that LoRA improves compression performance in the smallest 125m model, but not in larger models. We hypothesise that larger models are more prone to overfitting on the relatively few batches of wikitext-2 data used to compress the model. Nevertheless, we conclude that interleaved LoRA can be useful in cases, and recommend first using the proposed method without interleaved updates and, if enough data is available for compression, optionally using it if it improves performance.

Table 2: Structured compression of OPT-125m on wikitext-2 using interleaved LoRA updates

|  | Target Size | without LoRA | with LoRA |
|---|---|---|---|
| Pretrained | 100% | 27.65 | 23.35 |
| LLM Surgeon | 90% | 28.01 | 24.16 |
| (**ours**) | 80% | 28.73 | 25.25 |
| $G \otimes A$ | 70% | 31.82 | 28.86 |
| full cor. $\Delta$ | 60% | 38.47 | 31.26 |
|  | 50% | 49.78 | 36.50 |

## 4.3 Semi-structured Compression

For 2:4 semi-structured pruning, we compare LLM Surgeon with magnitude pruning, which only uses weight magnitudes, single-shot L-OBD, which only uses activations, and single-shot K-OBD, which also uses Kronecker-factored curvature but assumes full independence and thus only prunes without updating remaining weights as well as the recent state-of-the-art SparseGPT (Frantar & Alistarh, 2023). We report test performance after 50 % (2:4) semi-structured compression on wikitext-2 data in table 3. We empirically find that considering more weight correlations results in improved final performance after compression. Our proposed LLM Surgeon is competitive with prior work outperforming all baselines in terms of test set perplexity (PPL).

Table 3: Semi-structured 2:4 compression for large language models on wikitext-2 data.

| Method | $F \approx$ | Target size | Test performance (PPL) | | | |
|---|---|---|---|---|---|---|
|  |  |  | OPT (125m) | OPT (1.3b) | OPT (2.7b) | OPT (6.7b) |
| Baseline |  | 100% | **27.65** | **14.62** | **12.47** | **10.86** |
| Magnitude | $I \otimes I$ | 50% | 342.04 | 379.57 | 1106.01 | 187.29 |
| L-OBD | diag$(I \otimes A)$ | 50% | 87.26 | 44.92 | 41.40 | 27.36 |
| K-OBD | diag$(G \otimes A)$ | 50% | 68.74 | 27.22 | 20.23 | 15.55 |
| SparseGPT | $I \otimes A$ | 50% | 45.51 | 29.44 | 14.92 | 13.01 |
| LLM Surgeon (**ours**) | $G \otimes A$ | 50% | 44.64 | 25.10 | 14.64 | 12.10 |

## 4.4 Unstructured Compression

For unstructured pruning, we repeat the same experiments as structured pruning case described in section 4.1. In table 4, we report final test performance in terms of perplexity (PPL) on wikitext-2 after compressing LLMs of different sizes of OPT and Llama-v2 family. Overall, we find that methods with more accurate approximations of the curvature landscape and that account for more correlations perform better. The proposed LLM Surgeon outperforms all baselines, reaching the highest test performance across target sizes.

Table 4: Unstructured compression of large language models on wikitext-2 data.

| Method | Target size | Test performance (PPL) | | | | |
|---|---|---|---|---|---|---|
| | | OPT (125m) | OPT (1.3b) | OPT (2.7b) | OPT (6.7b) | Llama-v2 (7b) |
| Baseline | 100% | **27.65** | **14.62** | **12.47** | **10.86** | **5.12** |
| Magnitude | 90% | 27.62 | 14.69 | 12.60 | 10.88 | 5.18 |
| $I \otimes I$ | 80% | 28.53 | 15.68 | 13.18 | 11.26 | 5.37 |
| | 70% | 52.88 | 140.2 | 15.22 | 12.22 | 6.03 |
| L-OBD | 90% | 29.70 | 16.24 | 14.44 | 13.43 | 6.09 |
| $\text{diag}(I \otimes A)$ | 80% | 32.18 | 21.92 | 23.35 | 39.85 | 116.2 |
| single shot | 70% | 49.08 | 204.7 | 274.8 | 810.4 | 6549 |
| K-OBD | 90% | 27.64 | 14.62 | 12.09 | 36.89 | 5.13 |
| $G \otimes A$ | 80% | 27.62 | 14.37 | 130220 | 39928 | 5.19 |
| single shot | 70% | 27.92 | 220.1 | 23097 | 19506 | 5.60 |
| | 60% | 29.24 | 13783 | 10331 | 33896 | 9.20 |
| | 50% | 34.43 | 7311 | 10495 | 91506 | 118.6 |
| SparseGPT | 90% | 27.93 | 14.69 | 12.00 | 10.86 | 5.49 |
| $I \otimes A$ | 80% | 28.18 | 15.07 | 12.05 | 10.86 | 5.58 |
| | 70% | 28.93 | 22.77 | 12.17 | 10.89 | 5.71 |
| | 60% | 30.20 | 25.07 | 12.37 | 10.98 | 5.94 |
| | 50% | 33.17 | 26.77 | 12.88 | 11.92 | 6.51 |
| LLM Surgeon (**ours**) | 90% | 27.69 | 14.62 | 12.01 | 10.86 | 5.13 |
| $G_1 \otimes A_1$ | 80% | 27.83 | 14.66 | 12.14 | 10.87 | 5.20 |
| full cor. $\Delta$ | 70% | 28.35 | 14.81 | 12.25 | 10.82 | 5.36 |
| multi shot | 60% | 28.98 | 14.91 | 12.28 | 10.83 | 5.66 |
| | 50% | 30.30 | 15.47 | 12.68 | 10.97 | 6.08 |

## 4.5 LEARNED SPARSITY STRUCTURE

The proposed method can dynamically allocate sparsity across layers through global thresholds described in section 3.3. In Fig. 4.5, we compare total allocated sparsity levels per layer depth and per layer type after compressing a pretrained OPT-125m model. We find that the LLM Surgeon prunes relatively more in the first layer and less in middle layers. Further, we observe that a larger portions of weights are removed in fully-connected compared to attention blocks, but deviations are less compared to other methods. Dynamic allocation allows for most pruning where it hurts least.

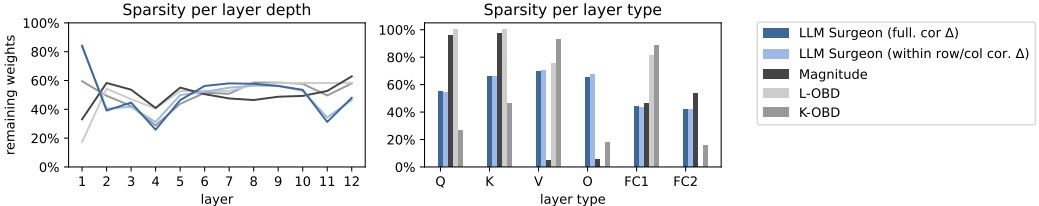

Figure 4: Sparsity levels obtained with structured pruning on OPT-125m by layer depth and type.

## 5 CONCLUSION

In this work, we have introduced the **LLM Surgeon** algorithm for unstructured, semi-structured and structured compression of neural networks. The work builds upon classic neural network compression approaches originating from the early 1990's that aim to find optimal pruning by expanding the curvature of the loss landscape. The method utilises modern Fisher approximations to scale accurate pruning to the realm of large language models (LLMs) with billions of parameters, while remaining practical in both memory and compute. Unlike most prior work on data-based LLM compression, we not only use weight magnitude and activations from forward passes, but also use gradient information from backward passes to relate weight removal costs to the true final objective. We improve upon prior work through more accurate approximations to the loss landscape curvature and considering more weight correlations to update remaining weights. Increasing the number of correlations and using multiple shots allows us trading off additional compute for better accuracy. Lastly, LLM Surgeon gives the first practically usable results for structured pruning of LLMs and achieves state-of-the-art results in unstructured and semi-structured large language model pruning.

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
