## A   DERIVATIONS FOR PRUNING

Given that we use a Gaussian approximation of our loss $p \approx q = \mathcal{N}$ through a quadratic approximation of our log likelihood $-\log p \approx \frac{1}{2}(\boldsymbol{\theta}^*)^T \boldsymbol{F} \boldsymbol{\theta}^*$, the most optimal compression becomes the solution to the following constrained optimization problem:

$$\underset{\Delta\boldsymbol{\theta}^*}{\arg\min} \; \frac{1}{2}\Delta(\boldsymbol{\theta}^*)^T \boldsymbol{F} \Delta\boldsymbol{\theta}^* \tag{14}$$

$$\text{s.t. } \boldsymbol{e}_k^T \Delta\boldsymbol{\theta}^* + \boldsymbol{e}_k^T \boldsymbol{\theta}^* = 0, \forall k \in Q$$

where $\mathcal{Q}$ is the set of $Q$ indices that are pruned.

### A.1   GENERAL SOLUTION

Following (Kurtic et al., 2022), we denote pruned elements as $\boldsymbol{E}_K = \begin{bmatrix} \boldsymbol{e}_{q_1} & \boldsymbol{e}_{q_2} & \dots \end{bmatrix}^T \in [0,1]^{|Q| \times P}$ and use the fact that solving eq. (6) through use of Langrange multipliers gives the general closed-form solution for cost $\mathcal{L}$ and weight update $\Delta\boldsymbol{\theta}$:

$$\mathcal{L} = \frac{1}{2}(\boldsymbol{E}_K \boldsymbol{\theta}^*)^T \left(\boldsymbol{E}_K \boldsymbol{F}^{-1} \boldsymbol{E}_K^T\right)^{-1} \boldsymbol{E}_K \boldsymbol{\theta}^* \tag{15}$$

$$\Delta\boldsymbol{\theta}^* = \boldsymbol{F}^{-1} \boldsymbol{E}_K^T \left(\boldsymbol{E}_K \boldsymbol{F}^{-1} \boldsymbol{E}_K^T\right)^{-1} \boldsymbol{E}_K \boldsymbol{\theta}^* \tag{16}$$

### A.2   REMOVING A SINGLE ELEMENT

**Optimal brain surgeon (OBS)**    To remove a single element with index $q$, we simply set $\boldsymbol{E}_K = \boldsymbol{e}_k^T$:

$$\begin{aligned} \mathcal{L} &= \frac{1}{2}(\boldsymbol{E}_K \boldsymbol{\theta}^*)^T \left(\boldsymbol{E}_K \boldsymbol{F}^{-1} \boldsymbol{E}_K^T\right)^{-1} \boldsymbol{E}_K \boldsymbol{\theta} \\ &= \frac{1}{2}\boldsymbol{\theta}_k^T \frac{1}{[\boldsymbol{F}^{-1}]_{kk}} \boldsymbol{\theta}_k \\ &= \frac{1}{2} \frac{(\boldsymbol{\theta}_k)^2}{[\boldsymbol{F}^{-1}]_{kk}} \end{aligned} \quad , \quad \begin{aligned} \Delta\boldsymbol{\theta} &= -\boldsymbol{F}^{-1} \boldsymbol{E}_K^T \left(\boldsymbol{E}_K \boldsymbol{F}^{-1} \boldsymbol{E}_K^T\right)^{-1} \boldsymbol{E}_K \boldsymbol{\theta} \\ &= -\boldsymbol{F}^{-1} \boldsymbol{e}_k \left(\boldsymbol{e}_k^T \boldsymbol{F}^{-1} \boldsymbol{e}_k\right)^{-1} \boldsymbol{e}_K^T \boldsymbol{\theta} \\ &= -\frac{\boldsymbol{\theta}_k}{[\boldsymbol{F}^{-1}]_{kk}} \boldsymbol{F}^{-1} \boldsymbol{e}_k \end{aligned} \tag{17}$$

which exactly correspond to the loss and updates of *optimal brain surgeon* (Hassibi & Stork, 1992).

**Optimal brain damage (OBD)**    We may also consider that elements are independent and the Fisher is diagonal. After noting that this implies that diagonal elements of the inverse Fisher are scalar inverses of elements in the Fisher $[\boldsymbol{F}^{-1}]_{kk} = \frac{1}{[\boldsymbol{F}]_{kk}}$, the formula's simplify to:

$$\mathcal{L} = [\boldsymbol{F}]_{kk}(\boldsymbol{\theta}_k)^2, \qquad\qquad \Delta\boldsymbol{\theta} = -\boldsymbol{\theta}_k \boldsymbol{e}_k \tag{18}$$

which exactly corresponds to loss and updates of *optimal brain damage* (LeCun et al., 1989).

### VECTORISED

For implementation purposes, it might be convenient to have a vectorised notation $\mathcal{L}_{\boldsymbol{\theta}} \in \mathbb{R}^{RC}$ or $\mathcal{L}_{\boldsymbol{W}} \in \mathbb{R}^{R \times C}$ to calculate all expected losses in parallel:

$$\begin{aligned} \text{For OBD:} \quad & \mathcal{L}_{\boldsymbol{\theta}} = \frac{1}{2}\boldsymbol{\theta}^* \odot \boldsymbol{\theta}^* \odot \text{diag}(\boldsymbol{F}) & \mathcal{L}_{\boldsymbol{W}} = \frac{1}{2}\boldsymbol{W}^* \odot \boldsymbol{W}^* \odot \text{mat}(\text{diag}(\boldsymbol{F})) \\ \text{For OBS:} \quad & \mathcal{L}_{\boldsymbol{\theta}} = \frac{1}{2}\boldsymbol{\theta}^* \odot \boldsymbol{\theta}^* \oslash \text{diag}(\boldsymbol{F}^{-1}) & \mathcal{L}_{\boldsymbol{W}} = \frac{1}{2}\boldsymbol{W}^* \odot \boldsymbol{W}^* \oslash \text{mat}(\text{diag}(\boldsymbol{F}^{-1})) \end{aligned} \tag{19}$$

### A.3   REMOVING A SINGLE ROW OR COLUMN

**Structured OBS**    If we consider the approximation $\boldsymbol{F} \approx \boldsymbol{G} \otimes \boldsymbol{A}$ with known inverse $(\boldsymbol{G} \otimes \boldsymbol{A})^{-1} = \boldsymbol{G}^{-1} \otimes \boldsymbol{A}^{-1}$, then to remove a row at index $r \in [0, R]$, we must take into account correlations within

elements of that row. That is, we write matrix $\boldsymbol{E}_K = (e_r^T \otimes \boldsymbol{I})$ containing one-hot row-vectors for all elements in row $r$. Plugging into the general solution eq. (7), we find:

$$
\begin{aligned}
\mathcal{L} &= \frac{1}{2} \boldsymbol{E}_K \boldsymbol{\theta}^T \left( \boldsymbol{E}_K \boldsymbol{F}^{-1} \boldsymbol{E}_K^T \right)^{-1} \boldsymbol{E}_K \boldsymbol{\theta}^* \\
&= \frac{1}{2} ((e_r^T \otimes \boldsymbol{I}) \boldsymbol{\theta}^*)^T \left( (e_r^T \otimes \boldsymbol{I})(\boldsymbol{G} \otimes \boldsymbol{A})^{-1}(e_r^T \otimes \boldsymbol{I})^T \right)^{-1} (e_r^T \otimes \boldsymbol{I}) \boldsymbol{\theta}^* \\
&= \frac{1}{2} \boldsymbol{\theta}_r^T \left( e_r^T \boldsymbol{G}^{-1} e_r \otimes \boldsymbol{I} \boldsymbol{A}^{-1} \boldsymbol{I} \right)^{-1} \boldsymbol{\theta}_r \\
&= \frac{1}{2} \boldsymbol{\theta}^T (e_r^T \otimes \boldsymbol{I}) \left( [[\boldsymbol{G}^{-1}]_{rr}] \otimes \boldsymbol{A}^{-1} \right)^{-1} (e_r \otimes \boldsymbol{I}) \boldsymbol{\theta}_r \\
&= \frac{1}{2} \frac{\boldsymbol{\theta}_r^T \boldsymbol{A} \boldsymbol{\theta}_r}{[\boldsymbol{G}^{-1}]_{rr}}
\end{aligned}
\tag{20}
$$

where we write $\boldsymbol{\theta}_r = e_r^T \boldsymbol{W}^* \in \mathbb{R}^C$ for the $r$'th row-vector in $\boldsymbol{W}$. Similarly, we obtain the associated weight update:

$$
\begin{aligned}
\Delta \boldsymbol{\theta} &= -\boldsymbol{F}^{-1} \boldsymbol{E}_K^T \left( \boldsymbol{E}_K \boldsymbol{F}^{-1} \boldsymbol{E}_K^T \right)^{-1} \boldsymbol{E}_K \boldsymbol{\theta}^* \\
&= -(\boldsymbol{G} \otimes \boldsymbol{A})^{-1} (e_r^T \otimes \boldsymbol{I})^T \left( (e_r^T \otimes \boldsymbol{I}) (\boldsymbol{G} \otimes \boldsymbol{A})^{-1} (e_r^T \otimes \boldsymbol{I})^T \right)^{-1} (e_r^T \otimes \boldsymbol{I}) \boldsymbol{\theta}^* \\
&= -(\boldsymbol{G}^{-1} \otimes \boldsymbol{A}^{-1}) (e_r \otimes \boldsymbol{I}) \left( e_r^T \boldsymbol{G}^{-1} e_r \otimes \boldsymbol{A}^{-1} \right)^{-1} \boldsymbol{\theta}_r \\
&= -\frac{1}{[\boldsymbol{G}^{-1}]_{rr}} \left( \boldsymbol{G}^{-1} e_r \otimes \boldsymbol{A}^{-1} \boldsymbol{I} \boldsymbol{A}^{-1} \boldsymbol{I} \right) \boldsymbol{\theta}_r \\
&= -\frac{\boldsymbol{G}^{-1} e_r \otimes \boldsymbol{\theta}_r}{[\boldsymbol{G}^{-1}]_{rr}}
\end{aligned}
\tag{21}
$$

arriving at a similar structured pruning update as derived in (Wang et al., 2019) for convolutional filters. We can equivalently derive expected loss and update for columns, by considering $\boldsymbol{E}_K = (\boldsymbol{I} \otimes e_c^T)$. If we do so, we find the structured updates for a row $r$ or column $c$:

$$
\begin{array}{lll}
\text{Remove row } r: & \mathcal{L} = \frac{1}{2} \dfrac{\boldsymbol{\theta}_r^T \boldsymbol{A} \boldsymbol{\theta}_r}{[\boldsymbol{G}^{-1}]_{rr}} & \Delta \boldsymbol{\theta} = -\dfrac{\boldsymbol{G}^{-1} e_r \otimes \boldsymbol{\theta}_r}{[\boldsymbol{G}^{-1}]_{rr}} \\[2ex]
\text{Remove column } c: & \mathcal{L} = \frac{1}{2} \dfrac{\boldsymbol{\theta}_c^T \boldsymbol{G} \boldsymbol{\theta}_c}{[\boldsymbol{A}^{-1}]_{cc}} & \Delta \boldsymbol{\theta} = -\dfrac{\boldsymbol{\theta}_c \otimes \boldsymbol{A}^{-1} e_c}{[\boldsymbol{A}^{-1}]_{cc}}
\end{array}
\tag{22}
$$

**Structured OBD** We may also assume that, when removing a row $r$, the individual elements within the row are also independent which would imply $[\boldsymbol{A}]_{ii} = \frac{1}{[\boldsymbol{A}^{-1}]_{ii}}$. Similarly, $[\boldsymbol{G}]_{ii} = \frac{1}{[\boldsymbol{G}^{-1}]_{ii}}$ when removing a column $c$. Consequently, we can simplify to:

$$
\begin{array}{lll}
\text{Remove row } r: & \mathcal{L} = \frac{1}{2} \boldsymbol{G}_{rr} \boldsymbol{\theta}_r^T \boldsymbol{A} \boldsymbol{\theta}_r & \Delta \boldsymbol{\theta} = -e_r \boldsymbol{\theta}_r^T \\[2ex]
\text{Remove column } c: & \mathcal{L} = \frac{1}{2} \boldsymbol{A}_{cc} \boldsymbol{\theta}_c^T \boldsymbol{G} \boldsymbol{\theta}_c & \Delta \boldsymbol{\theta} = -\boldsymbol{\theta}_c e_c^T
\end{array}
\tag{23}
$$

similar form to structured OBD losses and updates as derived in (Wang et al., 2019) for convolutional filters. The derivations slightly differ in that we start from the general solution eq. (8), circumventing the need to rederive a Langrange multipliers for each possible structure.

A.4 PRUNING MULTIPLE (CORRELATED) ROWS AND COLUMNS

Let us consider the removal of $R'$ rows $r_1, r_2, \ldots r'_R$ rows or $C'$ columns with indices $c_1, c_2, \ldots, c_{C'}$, with $1 < R' < R$ and $1 < C' < C$. We denote matrices containing one-hot vectors selecting all rows and columns to be removed respectively as:

$$
\boldsymbol{E}_{R'} = \begin{bmatrix} e_1 & e_2 & \ldots & e_{R'} \end{bmatrix}^T \in \mathbb{R}^{R' \times R} \qquad \boldsymbol{E}_{C'} = \begin{bmatrix} e_1 & e_2 & \ldots & e_{C'} \end{bmatrix}^T \in \mathbb{R}^{C' \times C}
\tag{24}
$$

Then, the matrix $\boldsymbol{E}_K$ containing one-hot row vectors selecting all elements to be removed can be written as:

$$
\begin{array}{lll}
\text{Multiple rows:} & \boldsymbol{E}_K = (\boldsymbol{E}_{R'} \otimes \boldsymbol{I}_C) \in \mathbb{R}^{Q \times RC}, \text{ (with } Q = R'C) \\[1ex]
\text{Multiple columns:} & \boldsymbol{E}_K = (\boldsymbol{I}_R \otimes \boldsymbol{E}_{C'}) \in \mathbb{R}^{Q \times RC}, \text{ (with } Q = RC')
\end{array}
\tag{25}
$$

To simultaneously remove rows and columns, we can stack the matrices with duplicate row vectors removed:

$$\text{Multiple rows and columns:} \quad \boldsymbol{E}_K \begin{bmatrix} \boldsymbol{E}_{R'} \otimes \boldsymbol{I}_C \\ \boldsymbol{I}_R \otimes \boldsymbol{E}_{C'} \end{bmatrix} \in \mathbb{R}^{Q \times RC} \text{ with duplicate rows removed} \quad (26)$$

The removal of duplicate rows is required due to the few $R'C'$ overlapping elements between rows and columns, after which the total number of rows thus becomes $Q = R'C + C'R - R'C'$. We used appropriately sized identity matrices $\boldsymbol{I}_R \in \mathbb{R}^{R \times R}$ and $\boldsymbol{I}_C \in \mathbb{R}^{C \times C}$. For brevity, we write the vector or matrix of pruned weights $\bar{\boldsymbol{\theta}} := \boldsymbol{E}_K \boldsymbol{\theta} \in \mathbb{R}^Q$.

First, we derive the removal for $R'$ rows by defining removal matrix as $\boldsymbol{E}_K = \boldsymbol{E}_{R'} \otimes \boldsymbol{I}$ and define $\overline{\boldsymbol{W}} := \boldsymbol{E}_{R'} \boldsymbol{W} \in \mathbb{R}^{R' \times C}$. The complete weight update for the removal of multiple rows becomes:

$$\begin{aligned}
\Delta \boldsymbol{\theta} &= -\boldsymbol{F}^{-1} \boldsymbol{E}_K^T \left( \boldsymbol{E}_K \boldsymbol{F}^{-1} \boldsymbol{E}_K^T \right)^{-1} \boldsymbol{E}_K \boldsymbol{\theta}^* \\
&= -(\boldsymbol{G} \otimes \boldsymbol{A})^{-1} (\boldsymbol{E}_{R'} \otimes \boldsymbol{I})^T \left( (\boldsymbol{E}_{R'} \otimes \boldsymbol{I})(\boldsymbol{G} \otimes \boldsymbol{A})^{-1} (\boldsymbol{E}_{R'} \otimes \boldsymbol{I})^T \right)^{-1} (\boldsymbol{E}_{R'} \otimes \boldsymbol{I}) \boldsymbol{\theta}^* \\
&= -(\boldsymbol{G}^{-1} \boldsymbol{E}_{R'}^T \otimes \boldsymbol{A}^{-1}) \left( \boldsymbol{E}_{R'} \boldsymbol{G}^{-1} \boldsymbol{E}_{R'}^T \otimes \boldsymbol{A}^{-1} \right)^{-1} \overline{\boldsymbol{\theta}^*} \\
&= -(\boldsymbol{G}^{-1} \boldsymbol{E}_{R'}^T \otimes \boldsymbol{A}^{-1}) \left( (\boldsymbol{E}_{R'} \boldsymbol{G}^{-1} \boldsymbol{E}_{R'}^T)^{-1} \otimes \boldsymbol{A} \right) \overline{\boldsymbol{\theta}^*} \\
\Delta \boldsymbol{W} &= -\boldsymbol{G}^{-1} \boldsymbol{E}_{R'}^T \left( (\boldsymbol{E}_{R'} \boldsymbol{G}^{-1} \boldsymbol{E}_{R'}^T)^{-1} \overline{\boldsymbol{W}} \boldsymbol{A} \right) \boldsymbol{A}^{-1} \\
&= -\boldsymbol{G}^{-1} \boldsymbol{E}_{R'}^T (\boldsymbol{E}_{R'} \boldsymbol{G}^{-1} \boldsymbol{E}_{R'}^T)^{-1} \overline{\boldsymbol{W}} \quad (27)
\end{aligned}$$

Similarly, we derive the removal of $C'$ columns by defining removal matrix as $\boldsymbol{E}_K = \boldsymbol{I} \otimes \boldsymbol{E}_{C'}$ and define $\overline{\boldsymbol{W}} := \boldsymbol{E}_{C'} \boldsymbol{W} \in \mathbb{R}^{R \times C'}$. The complete weight update for multiple column removal becomes:

$$\begin{aligned}
\Delta \boldsymbol{\theta} &= -\boldsymbol{F}^{-1} \boldsymbol{E}_K^T \left( \boldsymbol{E}_K \boldsymbol{F}^{-1} \boldsymbol{E}_K^T \right)^{-1} \boldsymbol{E}_K \boldsymbol{\theta}^* \\
&= -(\boldsymbol{G} \otimes \boldsymbol{A})^{-1} (\boldsymbol{I} \otimes \boldsymbol{E}_{C'}))^T \left( (\boldsymbol{I} \otimes \boldsymbol{E}_{C'})(\boldsymbol{G} \otimes \boldsymbol{A})^{-1} (\boldsymbol{I} \otimes \boldsymbol{E}_{C'})^T \right)^{-1} (\boldsymbol{I} \otimes \boldsymbol{E}_{C'}) \boldsymbol{\theta}^* \\
&= -(\boldsymbol{G} \otimes \boldsymbol{A})^{-1} (\boldsymbol{I} \otimes \boldsymbol{E}_{C'}))^T \left( (\boldsymbol{I} \otimes \boldsymbol{E}_{C'})(\boldsymbol{G} \otimes \boldsymbol{A})^{-1} (\boldsymbol{I} \otimes \boldsymbol{E}_{C'})^T \right)^{-1} (\boldsymbol{I} \otimes \boldsymbol{E}_{C'}) \boldsymbol{\theta}^* \\
&= -(\boldsymbol{G}^{-1} \otimes \boldsymbol{A}^{-1} \boldsymbol{E}_{C'}^T) \left( \boldsymbol{G} \otimes \boldsymbol{E}_{C'} \boldsymbol{A}^{-1} \boldsymbol{E}_{C'}^T \right)^{-1} \bar{\boldsymbol{\theta}} \\
\Delta \boldsymbol{W} &= -\boldsymbol{G}^{-1} \boldsymbol{G} \overline{\boldsymbol{W}} (\boldsymbol{E}_{C'} \boldsymbol{A}^{-1} \boldsymbol{E}_{C'}^T)^{-1} (\boldsymbol{A}^{-1} \boldsymbol{E}_{C'}^T) \\
&= -\overline{\boldsymbol{W}} (\boldsymbol{E}_{C'} \boldsymbol{A}^{-1} \boldsymbol{E}_{C'}^T)^{-1} (\boldsymbol{A}^{-1} \boldsymbol{E}_{C'}^T) \quad (28)
\end{aligned}$$

# B  EXPERIMENTAL DETAILS.

Code is available at: https://github.com/Qualcomm-AI-research/llm-surgeon.

## B.1  MODELS

**OPT models**   From the OPT model family ((Zhang et al., 2022)), we consider models with the following number of parameters: 125 million (125m), 1.3 billion (1.3b), 2.7 billion (2.7b), 6.7 billion (6.7b) models. We omit 350 million model due to different layer norm. We obtain the standard pre-trained checkpoints using Huggingface (Wolf et al., 2019) and use this as a baseline and initialisation for compression.

**Llama-v2 models**   From the Llama-v2 model family ((Touvron et al., 2023)), we consider a model with 7 billion (7b) parameters and a model with 13 billion (13b) parameters. We obtain the standard pre-trained checkpoints using Huggingface (Wolf et al., 2019) and use this as a baseline and initialisation for compression.

## B.2  DATASETS

**English / Wikitext-2**   The majority of the results are obtained on the Wikitext-2 dataset containing parsed subsets of the English Wikipedia (Merity et al., 2016; Wikipedia, 2004), using the default training and test sets. For fitting, we use 128 batches of 2048 characters and for testing we use the standard test set containing 4358 characters.

**French / Wikipedia**   For French data experiments, we use a subset of French wikipedia (Wikipedia, 2004). For fitting, we use 128 batches of 2048 characters and for testing we use a randomly selected test set containing 1067888 characters.

**German / Wikipedia**   For the Italian data experiments, we use a subset of the German wikipedia (Wikipedia, 2004). For fitting, we use 128 batches of 2048 characters and for testing we use a randomly selected test set containing 1112372 characters.

**Italian / Wikipedia**   For the Italian data experiments, we use a subset of the Italian wikipedia (Wikipedia, 2004). For fitting, we use 128 batches of 2048 characters and for testing we use a randomly selected test set containing 633177 characters.

## B.3  MASK EQUIVALENCE

When comparing the equivalence of obtained pruning masks between two models $\boldsymbol{\theta}_A$ and $\boldsymbol{\theta}_B$ obtained by two compression methods $A$ and $B$. We always consider the case of 50% pruning, and define the mask equivalence as the fraction of same weights that are set two zero in both models:

$$\text{mask equivalence} = \sum_{i=1}^{P} \frac{\mathbf{1}([\boldsymbol{\theta}_A]_i = 0 \text{ and } [\boldsymbol{\theta}_B]_i = 0)}{P}. \tag{29}$$

where $\mathbf{1}$ denotes an indicator function that returns 1 if both weights $[\boldsymbol{\theta}_A]_i$ and $[\boldsymbol{\theta}_B]_i$ are zero, and returns 0 otherwise.

## B.4  SPARSEGPT AND EVALUATION OF BASELINES

For the SparseGPT baseline, we used the official code SparseGPT code repository (Frantar & Alistarh, 2023) which allows for training and evaluation on wikitext-2. The obtained results may differ from those reported in the original paper as the C4 dataset was used there.

In this work, models were trained with the same 128 batches of the wikitext-2 training set as available in the SparseGPT codebase and are evaluated on the wikitext-2 test set using the exact same evaluation procedure.

## C  TECHNICAL DETAILS

### C.1  PSEUDOCODES

---

**Algorithm 2** LLM Surgeon (*structured*)

---

**Input:** target size $\alpha$
**Input:** initial weights $\boldsymbol{\theta}^0$
    **For** shot $t$ in $[1, 2, \ldots, T]$
        **Compute:** approximate curvature $\boldsymbol{G}_1, \boldsymbol{A}_1$ from data (optionally also $\boldsymbol{G}_2, \boldsymbol{A}_2$)  ▷ section 3.1
        **Compute:** costs per row/column $\mathcal{L}_r, \mathcal{L}_c$ from $\boldsymbol{G}_1, \boldsymbol{A}_1, (\boldsymbol{G}_2, \boldsymbol{A}_2)$  ▷ section 3.2
        **Compute:** threshold $\tau$ using $\mathcal{L}_r$ and $\mathcal{L}_c$ given target size $\alpha$  ▷ section 3.3
        **Select:** rows and columns to remove $\boldsymbol{E}_R, \boldsymbol{E}_C$ based on $\tau$  ▷ section 3.3
        **Compute:** weight update $\Delta\boldsymbol{\theta}^{t-1}$ based on $\boldsymbol{E}_R, \boldsymbol{E}_C$ and $\boldsymbol{G}_1, \boldsymbol{A}_1, (\boldsymbol{G}_2, \boldsymbol{A}_2)$  ▷ section 3.4
        **Update:** remaining weights $\boldsymbol{\theta}^t \leftarrow \boldsymbol{\theta}^{t-1} + \Delta\boldsymbol{\theta}^{t-1}$  ▷ section 3.5
        **Optionally:** $\boldsymbol{\theta}^t \leftarrow$ low-rank update($\boldsymbol{\theta}^t$)
    **Output:** compressed weights $\hat{\boldsymbol{\theta}} = \boldsymbol{\theta}^T$

---

**Algorithm 3** LLM Surgeon (*semi-structured / unstructured*)

---

**Input:** target size $\alpha$
**Input:** initial weights $\boldsymbol{\theta}^0$
    **For** shot $t$ in $[1, 2, \ldots, T]$
        **Compute:** approximate curvature $\boldsymbol{G}_1, \boldsymbol{A}_1$ from data (optionally also $\boldsymbol{G}_2, \boldsymbol{A}_2$)  ▷ section 3.1
        **Compute:** costs per element $\mathcal{L}_k$ from $\boldsymbol{G}_1, \boldsymbol{A}_1, (\boldsymbol{G}_2, \boldsymbol{A}_2)$  ▷ section 3.2
        **Compute:** threshold $\tau$ from $\mathcal{L}_k$ and target size $\alpha_t$ (unstructured/semistructured)▷ section 3.3
        **Select:** elements to remove $\boldsymbol{E}_K$ based on $\tau$ (unstructured/semistructured)  ▷ section 3.3
        **Compute:** weight update $\Delta\boldsymbol{\theta}^{t-1}$ based on $\boldsymbol{E}_K$ and $\boldsymbol{G}_1, \boldsymbol{A}_1, (\boldsymbol{G}_2, \boldsymbol{A}_2)$  ▷ section 3.4
        **Update:** remaining weights $\boldsymbol{\theta}^t \leftarrow \boldsymbol{\theta}^{t-1} + \Delta\boldsymbol{\theta}^{t-1}$  ▷ section 3.5
        **Optionally:** $\boldsymbol{\theta}^t \leftarrow$ low-rank update($\boldsymbol{\theta}^t$)
    **Output:** compressed weights $\hat{\boldsymbol{\theta}} = \boldsymbol{\theta}^T$

---

### C.2  DAMPENING

In practice, we dampen the $\boldsymbol{G}$ and $\boldsymbol{A}$ matrices by adding a diagonal term $\boldsymbol{G} + \lambda_G \boldsymbol{I}$ and $\boldsymbol{A} + \lambda_A \boldsymbol{I}$. In our experiments, we found that values in the range $[0.01, 0.1]$ multiplied by mean diagonal terms generally works well. We follow (Frantar & Alistarh, 2023) and always use $\lambda_A{=}0.01\mathrm{diag}(\boldsymbol{A})$ to be consistent with prior work and allow for a fair comparison with baselines. Further, we use $\lambda_G{=}0.1\mathrm{diag}(\boldsymbol{G})$ for structured experiments and $\lambda_G{=}0.01\mathrm{diag}(\boldsymbol{G})$ in semi-structured and unstructured experiments.

## D DOWNSTREAM TASK PERFORMANCE

We also evaluate our method on downstream tasks as perplexity metrics do not necessarily correlate with downstream performance. Further, we also repeat this experiment using the C4 dataset as reference data for compression, as this is used in prior work (Frantar & Alistarh, 2023) and as this can be regarded a more general reference dataset. In tables 5 and 6 we report 0-shot test performance of structured pruning for LLM surgeon and K-OBD baseline.

Table 5: Downstream task performance using Wikitext-2 for pruning.

| Structured pruning (with wikitext-2) | Model size | wikitext word ppl | boolq | piqa | hallaswag | winogrande | arc_easy | arc_challenge | openbookq | copa | lambada_openai | wsc273 | AVERAGE wikitext2 |
|---|---|---|---|---|---|---|---|---|---|---|---|---|---|
| Dense baseline | 100% | 9.24 | 77.74 | 79.11 | 75.99 | 69.14 | 74.58 | 46.25 | 44.20 | 86.00 | 73.92 | 85.71 | 71.26 |
| LLM Surgeon (ours) | 90% | 9.63 | 76.21 | 78.56 | 75.39 | 67.64 | 74.12 | 46.50 | 43.60 | 85.00 | 72.64 | 84.98 | 70.46 |
|  | 80% | 12.16 | 72.97 | 77.09 | 71.30 | 66.30 | 71.36 | 41.89 | 41.80 | 87.00 | 56.43 | 80.22 | 66.66 |
|  | 70% | 16.91 | 61.25 | 73.56 | 60.72 | 61.09 | 63.09 | 36.69 | 38.80 | 81.00 | 28.33 | 76.56 | 58.11 |
|  | 60% | 25.15 | 44.98 | 69.26 | 48.04 | 54.38 | 52.31 | 30.29 | 36.80 | 78.00 | 11.72 | 68.50 | 49.43 |
|  | 50% | 43.68 | 39.60 | 64.36 | 40.29 | 52.57 | 44.91 | 26.28 | 30.80 | 74.00 | 6.52 | 61.54 | 44.09 |
| K-OBD | 90% | 9.89 | 76.67 | 78.02 | 74.80 | 68.11 | 75.17 | 46.33 | 44.60 | 86.00 | 72.71 | 82.78 | 70.52 |
|  | 80% | 17.62 | 74.34 | 75.24 | 67.85 | 64.64 | 63.80 | 40.27 | 41.60 | 83.00 | 30.23 | 82.42 | 62.34 |
|  | 70% | 32.72 | 65.29 | 71.82 | 53.07 | 56.83 | 51.05 | 33.11 | 37.80 | 79.00 | 12.21 | 70.70 | 53.09 |
|  | 60% | 68.63 | 60.80 | 65.67 | 43.99 | 53.20 | 41.79 | 28.50 | 34.00 | 75.00 | 7.04 | 60.44 | 47.04 |
|  | 50% | 136.33 | 61.56 | 60.66 | 36.84 | 53.04 | 36.11 | 26.71 | 33.00 | 72.00 | 4.70 | 61.17 | 44.58 |

Table 6: Downstream task performance using C4 for pruning.

| Structured pruning (with C4) | Model size | wikitext word ppl | boolq | piqa | hallaswag | winogrande | arc_easy | arc_challenge | openbookq | copa | lambada_openai | wsc273 | AVERAGE wikitext2 |
|---|---|---|---|---|---|---|---|---|---|---|---|---|---|
| Dense baseline | 100% | 9.24 | 77.74 | 79.11 | 75.99 | 69.14 | 74.58 | 46.25 | 44.20 | 86.00 | 73.92 | 85.71 | 71.26 |
| LLM Surgeon (ours) | 90% | 9.90 | 77.03 | 78.45 | 74.95 | 68.27 | 73.19 | 45.99 | 44.60 | 84.00 | 72.81 | 82.78 | 70.21 |
|  | 80% | 14.42 | 75.60 | 76.82 | 69.71 | 63.85 | 70.29 | 41.30 | 42.80 | 87.00 | 45.53 | 82.42 | 65.53 |
|  | 70% | 25.16 | 66.39 | 72.85 | 58.11 | 56.83 | 62.16 | 34.47 | 38.40 | 80.00 | 22.69 | 69.96 | 56.19 |
|  | 60% | 45.35 | 62.48 | 68.93 | 48.10 | 55.64 | 51.56 | 27.99 | 35.20 | 70.00 | 12.56 | 61.54 | 49.40 |
|  | 50% | 77.30 | 62.60 | 65.02 | 41.70 | 54.22 | 42.55 | 24.23 | 31.20 | 71.00 | 7.26 | 60.44 | 46.02 |
| K-OBD | 90% | 10.59 | 75.47 | 78.18 | 73.61 | 66.46 | 72.52 | 44.37 | 43.60 | 87.00 | 71.22 | 82.42 | 69.48 |
|  | 80% | 20.12 | 73.36 | 75.14 | 66.11 | 62.43 | 62.84 | 38.23 | 41.00 | 86.00 | 21.50 | 78.39 | 60.50 |
|  | 70% | 56.92 | 63.30 | 68.44 | 52.31 | 55.64 | 46.72 | 31.31 | 34.60 | 77.00 | 5.69 | 68.86 | 50.39 |
|  | 60% | 112.85 | 62.23 | 64.47 | 46.36 | 52.17 | 40.53 | 29.52 | 32.40 | 72.00 | 2.91 | 63.00 | 46.56 |
|  | 50% | 272.16 | 62.42 | 61.70 | 38.47 | 50.43 | 33.29 | 26.96 | 31.80 | 65.00 | 0.91 | 59.34 | 43.03 |

We find that our method not only performs well in terms of test perplexity but also correlates well with downstream performance, outperforming the baselines on these downstream tasks.

## E ADDITIONAL EXPERIMENTS ON LLAMA-V2 13B.

To assess performance on larger 13B parameter models, we also report structured compression on the Llama-v2 13B model and evaluate downstream task performance. Test perplexities (lower is better) can be found in table 7 below:

Table 7: Pruning Llama-v2 13B model.

|  | Baseline | Pruned model sizes |  |  |  |  |
|---|---|---|---|---|---|---|
|  | Dense 100% | 90% | 80% | 70% | 60% | 50% |
| K-OBD | 4.547 | 4.908 | 6.294 | 10.08 | 13.06 | 16.06 |
| LLM Surgeon | 4.547 | 4.692 | 5.286 | 6.207 | 7.245 | 9.428 |

as well as evaluated results on downstream benchmarks (higher is better) in table 8 below.

Table 8: Downstream task performance after pruning large Llama-v2 13B model.

| Llama-v2 13B | Model size | wikitext word ppl | boolq | piqa | hallaswag | winogrande | arc_easy | arc_challenge | openbookq | copa | lambada_openai | wsc273 | AVERAGE wikitext2 |
|---|---|---|---|---|---|---|---|---|---|---|---|---|---|
| Dense baseline | 100% | 8.23 | 80.52% | 80.52% | 79.38% | 72.14% | 77.53% | 49.23% | 45.20% | 90.00% | 76.77% | 89.38% | 74.07% |
| LLM Surgeon (ours) | 90% | 8.57 | 81.07% | 79.87% | 79.24% | 72.38% | 76.30% | 49.91% | 47.20% | 92.00% | 75.47% | 89.38% | 74.28% |
|  | 80% | 10.08 | 80.86% | 79.00% | 77.09% | 70.56% | 75.93% | 46.76% | 46.80% | 90.00% | 67.79% | 86.45% | 72.12% |
|  | 70% | 12.74 | 74.50% | 76.50% | 71.52% | 68.67% | 69.74% | 40.27% | 45.00% | 91.00% | 54.40% | 83.52% | 67.51% |
|  | 60% | 16.00 | 64.62% | 73.01% | 65.04% | 65.75% | 63.80% | 37.12% | 39.60% | 90.00% | 44.50% | 81.32% | 62.48% |
|  | 50% | 23.75 | 65.66% | 68.77% | 56.19% | 63.22% | 56.19% | 31.83% | 36.60% | 85.00% | 35.16% | 77.29% | 57.59% |
| K-OBD | 90% | 8.79 | 81.31% | 79.76% | 79.12% | 72.22% | 76.94% | 47.95% | 47.80% | 91.00% | 75.26% | 88.64% | 74.00% |
|  | 80% | 11.79 | 80.80% | 79.16% | 76.80% | 70.56% | 73.74% | 46.93% | 48.60% | 88.00% | 58.99% | 87.55% | 71.11% |
|  | 70% | 20.00 | 66.76% | 74.43% | 64.18% | 64.96% | 56.23% | 36.01% | 39.00% | 88.00% | 38.54% | 79.49% | 60.76% |
|  | 60% | 27.74 | 55.66% | 70.24% | 55.52% | 60.46% | 49.62% | 32.68% | 35.80% | 80.00% | 30.06% | 73.63% | 54.37% |
|  | 50% | 37.38 | 59.79% | 66.54% | 48.39% | 57.46% | 46.59% | 30.72% | 34.00% | 77.00% | 24.61% | 69.96% | 51.50% |

We find that LLM Surgeon also outperforms baselines on existing Llama-v2 13B models. We stress that these results are obtained on structured pruning of rows and columns, which are regarded the hardest and most constrained pruning structure. Yet, we can compress Llama 13B by 20% with less than 2% drop in downstream task performance. It also significantly outperforms the baseline for all compression rates, both in terms of test perplexity and downstream task performance.

# F ABLATIONS

## F.1 SHOTS

Table 9: Ablation of shot counts $T$ for structured LLM Surgeon compressing OPT-1.3b model.

| Target size | Shots $T$ | wikitext-2 PPL | Shots $T$ | wikitext-2 PPL | Shots $T$ | wikitext-2 PPL |
|---|---|---|---|---|---|---|
| 90% | 6 | 14.70 | 8 | 14.70 | 10 | 14.72 |
| 80% | 12 | 15.14 | 16 | 15.12 | 20 | 15.08 |
| 70% | 18 | 16.21 | 24 | 16.24 | 30 | 16.23 |
| 60% | 24 | 18.53 | 32 | 18.45 | 40 | 18.49 |
| 50% | 30 | 23.32 | 40 | 22.95 | 50 | **22.68** |

## F.2 TASK-SPECIFIC COMPRESSION

LLM Surgeon uses data to find a compressed model that has the least negative impact on final test performance. In this section, we explore the extent to which the method can use data to compress specifically to the task at hand. We do so by comparing test performance and equivalences between resulting pruning masks for different language modeling languages: English (EN/wikitext-2), French (FR) and Italian (IT) and the German (DE). We consider 50%

Table 10: Cross-task performance and mask equivalences of 50% compressed OPT-125m model using structured LLM Surgeon on language subsets.

| target | evaluation dataset | | | | mask equivalence (%) | | | |
|---|---|---|---|---|---|---|---|---|
| | EN | FR | DE | IT | EN | FR | DE | IT |
| Pretrained | 27.66 | 22.54 | 24.32 | 27.66 | | | | |
| EN | **47.46** | 172.9 | 181.1 | 169.1 | 1.00 | 0.74 | 0.70 | 0.72 |
| FR | 113.4 | **28.44** | 35.02 | 34.90 | 0.74 | 1.00 | 0.87 | 0.90 |
| DE | 142.1 | 35.15 | **27.49** | 38.49 | 0.70 | 0.87 | 1.00 | 0.87 |
| IT | 123.7 | 31.85 | 33.78 | **30.58** | 0.72 | 0.90 | 0.87 | 1.00 |

unstructured compression using LLM Surgeon with correlated weight updates. For each compressed model, we compare performance on all languages and compare the equivalences between resulting pruning masks (details in appendix B.3), and report results in table 10. Like other methods that use data for compression (Hassibi & Stork, 1992; Frantar & Alistarh, 2023; Wang et al., 2019), we expect to see some correlation between the data used for training and data with good test performance, which is reflected in both test performance and masks. It is important to note that the final performance after compression will depend on the quality of the used dataset for compression. Further, the experiment demonstrates that the method can be used for task-specific compression tailored towards the data used for compression and generalises to high test performance on the associated test data.

# G  ON FAIR COMPARISON

All results in this work (including the SparseGPT) were trained on Wikitext-2 for fair comparison. To do so, we used the same dataloader and evaluation script as the official SparseGPT repo and reran all SparseGPT results to be trained on Wikitext-2. In some cases, this resulted in better scores for the SparseGPT baseline compared to the C4-trained results reported in the original SparseGPT paper. Yet, we find that our method using improved curvature estimates still outperformed the baselines in terms of final test performance.

# H  COMPUTATIONAL PERFORMANCE

We report computational cost in terms of pruning time in table 11 and GPU memory in table 12.

Table 11: Time performance.

| Runtime | Network | Time | Test performance | | | | |
| --- | --- | --- | --- | --- | --- | --- | --- |
| | | | PPL 90% | PPL 80% | PPL 70% | PPL 60% | PPL 50% |
| Unstructured baseline (SparseGPT) | Llama-v2 7B | <5m | 5.49 | 5.58 | 5.71 | 5.94 | 6.51 |
| Unstructured LLM Surgeon (**ours**) | Llama-v2 7B | 2d8h16m | 5.13 | 5.20 | 5.36 | 5.66 | 6.08 |
| Structured baseline (K-OBD) | Llama-v2 7B | 16h58m | 5.48 | 9.14 | 15.43 | 28.03 | 46.64 |
| Structured LLM Surgeon (**ours**) | Llama-v2 7B | 17h08m | 5.25 | 6.18 | 7.83 | 10.39 | 15.38 |
| Structured baseline (K-OBD) | Llama-v2 13B | 1d6h5m | 4.908 | 6.294 | 10.08 | 13.06 | 16.06 |
| Structured LLM Surgeon (**ours**) | Llama-v2 13B | 1d9h26m | 4.692 | 5.286 | 6.207 | 7.245 | 9.428 |

Our method is most efficient for structured pruning, but it must be noted that engineering efforts may further improve speed for unstructured pruning. The focus of the paper is structured pruning, on which we achieve state-of-the-art compression rates. Importantly, compression of LLMs only needs to happen once after which a pruned model can be deployed infinitely many times without further cost. This motivates our method which takes longer to run but reaches better final test performance.

Table 12: Memory performance.

| Network | SparseGPT (baseline) | Unstructured LLM-Surgeon (ours) |
| --- | --- | --- |
| Llama-7B | <5m / 1 GPU (32GB) | 2d8h16m / 4xH100 80 GB |
| | K-OBD (baseline) | Structured LLM-Surgeon (ours) |
| Llama-7B | 16h58m / 4xH100 80 GB | 17h08m / 4xH100 80 GB |
| Llama-13B | 1d6h5m / 8xH100 80 GB | 1d9h26m / 8xH100 80 GB |

We argue that differences in the performance and the runtime of pruning methods can largely be attributed to underlying assumptions on correlations between weights. Notably, algorithms that consider few correlations, sometimes to the extent of completely disregarding all gradient information, can result in very fast pruning algorithms for unstructured and semi-structured pruning but are often not flexible enough to perform structured pruning of rows and columns. Examples of such lightweight algorithms for LLMs are (Sun et al., 2023) and SparseGPT (Frantar & Alistarh, 2023), as can also be observed from table 11. Our approach makes less strong assumptions on the curvature of the loss and as a result outperforms all baselines on all unstructured, semi-structured and structured pruning. Further, the improved curvature is also eligible for dynamic allocation of weight removal and improved correlated weight updates. In practice, we always recommend using our method for structured pruning. For unstructured and semi-structured pruning, we note an important trade-off between the desired final test accuracy and the available computational budget. Here, our proposed method can achieve the highest final model performance but requires more computational resources and takes longer to run. It should be noted that pruning only needs to happen once after which a model can be deployed infinitely many times this time, which dependent on the available computational resources can also legitimise spending additional pruning time even if this is much higher compared to other algorithms in relative terms. In absolute terms, the use of multiple large GPUs is common practice in the field of large language models and many more GPUs are typically used to train and deploy large language models. Moreover, the curvature approximation is naively amenable to data parallelism in case further speed-ups or larger models are required. We hope this provides context and emphasises the trade-off between performance and compute in practice.

## I  EXTENDING CURVATURE ESTIMATES

Instead of using a single Kronecker product, we might consider improving the approximation through a sum of multiple Kronecker factors:

$$\boldsymbol{F} \approx \widetilde{\boldsymbol{F}} = \boldsymbol{G}_1 \otimes \boldsymbol{A}_1 + \boldsymbol{G}_2 \otimes \boldsymbol{A}_2 \tag{30}$$

This last appendix deals with the question how one may computationally find such approximations and how to utilise them in the neural network pruning framework.

### I.1  NEAREST KRONECKER PRODUCT OR SUM OF KRONECKER PRODUCTS

Instead of assuming independence of activations and derivatives as in section 3.1, following the classic KFAC of (Martens & Grosse, 2015), we might want to find the *nearest Kronecker product* approximation $\boldsymbol{F} \approx \widetilde{\boldsymbol{G}} \otimes \widetilde{\boldsymbol{A}}$ that is closest to the Fisher in terms of the Frobenius norm:

$$\widetilde{\boldsymbol{G}}_l, \widetilde{\boldsymbol{A}}_l = \underset{\boldsymbol{G}_l, \boldsymbol{A}_l}{\arg\min} \, ||\boldsymbol{F}_l - \boldsymbol{G}_l \otimes \boldsymbol{A}_l||_F \tag{31}$$

Finding the nearest sum of Kronecker factors can be rephrased as a classic eigenvalue problem of finding the nearest rank-1 matrix. Golub & Van Loan (2013).

$$||\boldsymbol{F} - \tilde{\boldsymbol{G}} \otimes \tilde{\boldsymbol{A}}||_F \quad \equiv \quad ||\mathcal{R}(\boldsymbol{F}) - \text{vec}(\widetilde{\boldsymbol{G}})\text{vec}(\widetilde{\boldsymbol{A}})^T||_F \tag{32}$$

**Power method and deflation**   After considering the reshaping, we can use power iterations to solve for and find the nearest Kronecker factors $\boldsymbol{G}_1, \boldsymbol{A}_1 = \text{solve}(\boldsymbol{F})$.

Find with power iterations:

$$\widetilde{G}_1, \widetilde{A}_1 = \text{solve}(\boldsymbol{F}) = \underset{\boldsymbol{G}, \boldsymbol{A}}{\arg\min} \, ||\boldsymbol{F} - \boldsymbol{G} \otimes \boldsymbol{A}||_F$$

Deflation:

$$\widetilde{G}_r, \widetilde{A}_r = \text{solve}(\boldsymbol{F} - \sum\nolimits_{r'=1}^{r-1}(\widetilde{\boldsymbol{G}}_{r'} \otimes \widetilde{\boldsymbol{A}}_{r'}))$$

A more extensive description of the power method solve($\cdot$) can be found in algorithm 4. At the start of the algorithm, we initialise power iterations as vector with one's $\mathbf{1} = [1 \quad 1 \quad \ldots \quad 1]$. After each shot we can initialise the vector as the final estimate found during the previous shot.

---

**Algorithm 4** Kronecker power method. Finds $\widetilde{\boldsymbol{G}}, \widetilde{\boldsymbol{A}}$ nearest Kronecker product $||\boldsymbol{F} - \widetilde{\boldsymbol{G}} \otimes \widetilde{\boldsymbol{A}}||_F$.

---
**Input:**  Initialise $\widetilde{\boldsymbol{g}}^0 = \mathbf{1}, \widetilde{\boldsymbol{a}}^0 = \mathbf{1}$ (or using estimates of previous shot).
**Input:**  Set iterations $I$ (or $I=1$ if using estimates from previous shot)
**Output:**  $\widetilde{\boldsymbol{G}}, \widetilde{\boldsymbol{A}}$
   **for** iteration $i$ in $[1, 2, \ldots, I]$ **do**
      **Compute:** $\widetilde{\boldsymbol{g}}^i = \frac{\mathcal{R}(\widetilde{\boldsymbol{F}})\widetilde{\boldsymbol{a}}^{i-1}}{||\mathcal{R}(\widetilde{\boldsymbol{F}})\widetilde{\boldsymbol{a}}^{i-1}||_2}$ , with $\mathcal{R}(\widetilde{\boldsymbol{F}})\widetilde{\boldsymbol{a}}^{i-1} = \frac{1}{N}\sum_{n=1}^{N} \boldsymbol{a}_n^T \widetilde{\boldsymbol{A}}^{i-1} \boldsymbol{a}_n \text{vec}(\boldsymbol{g}_n \boldsymbol{g}_n^T)$
      **Compute:** $\widetilde{\boldsymbol{a}}^i = \frac{\mathcal{R}(\widetilde{\boldsymbol{F}})^T \widetilde{\boldsymbol{g}}^i}{||\mathcal{R}(\widetilde{\boldsymbol{F}})^T \widetilde{\boldsymbol{g}}^i||_2}$ , with $\mathcal{R}(\widetilde{\boldsymbol{F}})^T \widetilde{\boldsymbol{g}}^i = \frac{1}{N}\sum_{n=1}^{N} \boldsymbol{g}_n^T \widetilde{\boldsymbol{G}}^i \boldsymbol{g}_n \text{vec}(\boldsymbol{a}_n \boldsymbol{a}_n^T)$
      **Compute:** $\sigma^i = ||\widetilde{\boldsymbol{a}}^i||_2$
   **end for**
**Return:** $\widetilde{\boldsymbol{G}} = \sqrt{\sigma^i}\text{mat}(\widetilde{\boldsymbol{g}}), \widetilde{\boldsymbol{A}} = \sqrt{\sigma^i}\text{mat}(\widetilde{\boldsymbol{a}})$.

---

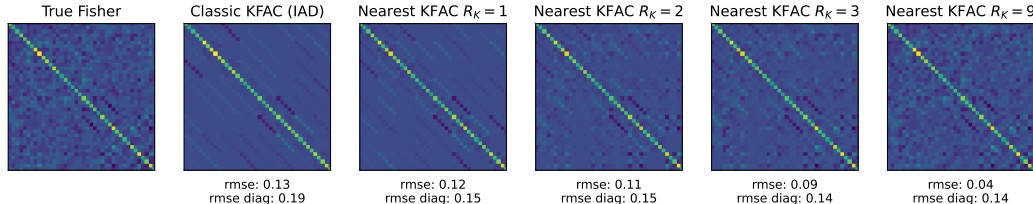

Figure 5: Example illustration of nearest Kronecker factor approximations $\widetilde{\boldsymbol{F}} \approx \sum_{r=1}^{R_K} \boldsymbol{G}_i \otimes \boldsymbol{A}_i$, compared to classical KFAC with the IAD assumption. Larger $R_K$ yields better approximations to the true Fisher $\boldsymbol{F}$ for larger $R_K$, as measured by the root mean squared error (rmse).

## I.2 EXTENDED CURVATURE APPROXIMATIONS

For classic KFAC with IAD or $R_K{=}1$ nearest Kronecker approximations of the form $\widetilde{\boldsymbol{F}} = \boldsymbol{G} \otimes \boldsymbol{A}$, the inverse simply becomes $(\boldsymbol{G} \otimes \boldsymbol{A})^{-1} = \boldsymbol{G}^{-1} \otimes \boldsymbol{A}^{-1}$. Unfortunately, we can not use this famous inverse identity for sum of Kronecker factors, which is why we fall back on eigendecompositions $\boldsymbol{G} = \boldsymbol{E}_1 \boldsymbol{S}_1 \boldsymbol{E}_1^T$ and $\boldsymbol{A} = \boldsymbol{E}_2 \boldsymbol{S}_2 \boldsymbol{E}_2^T$, allowing us to decompose the Fisher into:

$$\widetilde{\boldsymbol{F}} = \boldsymbol{K}_1 \boldsymbol{S}_1 \boldsymbol{K}_1^T \otimes \boldsymbol{K}_2 \boldsymbol{S}_2 \boldsymbol{K}_2^T = (\boldsymbol{K}_1 \otimes \boldsymbol{K}_2)(\boldsymbol{I} \otimes \boldsymbol{I} + \boldsymbol{S}_1 \otimes \boldsymbol{S}_2)(\boldsymbol{K}_1^T \otimes \boldsymbol{K}_2^T) \tag{33}$$

where specific $\boldsymbol{K}_1$ and $\boldsymbol{K}_2$ can be found in App. B of Martens & Grosse (2015), which we closely followed in our derivations. Because $\boldsymbol{K}_1$ and $\boldsymbol{K}_2$ are orthogonal and $\boldsymbol{S}_1$ and $\boldsymbol{S}_2$ diagonal, the inverse Fisher becomes:

$$\widetilde{\boldsymbol{F}}^{-1} = (\boldsymbol{K}_1 \otimes \boldsymbol{K}_2)(\boldsymbol{I} \otimes \boldsymbol{I} + \boldsymbol{S}_1 \otimes \boldsymbol{S}_2)^{-1}(\boldsymbol{K}_1^T \otimes \boldsymbol{K}_2^T) \tag{34}$$

In the context of neural network training, the problem gets slightly harder since we want to incrementally construct estimates $\widetilde{\boldsymbol{G}}_i$ and $\widetilde{\boldsymbol{A}}_i$ from individual samples $\boldsymbol{a}_{l,n}, \boldsymbol{g}_{l,n}$ that make up $\boldsymbol{F}$, without having to simultaneously store more than a single or batch of input activations $\boldsymbol{a}_{l,n}$ or output gradients $\boldsymbol{g}_{l,n}$ in memory. Although this *online Kronecker-product principal component analysis* problem largely remains an open research problem, we our approach closely follows the recent work by (Koroko et al., 2022) that uses similar approximations in the context of optimisation. A sum of multiple $R_K{>}1$ Kronecker factors will yield closer approximations, but also linearly increase memory requirements with higher $R_K$ and makes inverting $\boldsymbol{F}^{-1}$ considerably more difficult.

**Formulas to compute cost and weight updates.** For sum of Kronecker factors, we find that the constrained optimization solution of for costs $\Delta \mathcal{L}$ eq. (7) and weight updates $\Delta \boldsymbol{\theta}$ eq. (8) become the following inner-product and matrix-vector product:

$$\mathcal{L}_k = \frac{1}{2}\langle \overline{\boldsymbol{\theta}^*}, \boldsymbol{U}\overline{\boldsymbol{\theta}^*} \rangle = (\overline{\boldsymbol{\theta}^*})^T \boldsymbol{U}(\overline{\boldsymbol{\theta}^*}) \in \mathbb{R} \tag{35}$$

$$\Delta \boldsymbol{\theta} = \widetilde{\boldsymbol{F}}^{-1} \boldsymbol{E}_K^T \boldsymbol{u} = \boldsymbol{K}_1 \left( \overline{\boldsymbol{K}}_1^T \boldsymbol{U} \overline{\boldsymbol{K}}_2 \oslash \left[ \boldsymbol{1}\boldsymbol{1}^T + \boldsymbol{s}_1 \boldsymbol{s}_2^T \right] \right) \boldsymbol{K}_2^T \in \mathbb{R}^{RC} \tag{36}$$

with at the heart of it all a matrix $\boldsymbol{U} = [\boldsymbol{E}_K \boldsymbol{F}^{-1} \boldsymbol{E}_K^T]^{-1}$ that captures correlations between weights:

$$\boldsymbol{U} = \left[ \boldsymbol{E}_K \left( \boldsymbol{K}_1 \otimes \boldsymbol{K}_2 \right) \left( \boldsymbol{I} \otimes \boldsymbol{I} + \boldsymbol{S}_1 \otimes \boldsymbol{S}_2 \right)^{-1} \left( \boldsymbol{K}_1^T \otimes \boldsymbol{K}_2^T \right) \boldsymbol{E}_K^T \right]^{-1} \tag{37}$$

where $(\boldsymbol{I} \otimes \boldsymbol{I} + \boldsymbol{S}_1 \otimes \boldsymbol{S}_2)$ is diagonal and the inverse can thus be computed element-wise. The remaining inverse is of size $K \times K$, for $K$ correlated weights.

**Note on sum of Kronecker factors** Experimentally, we did not find a benefit in performance when using a sum of two nearest Kronecker factor approximation, or found it too slow. Therefore, we focus in the main text on LLM Surgeon with fast single Kronecker product KFAC approximation to approximate the loss landsscape curvature. Nevertheless, we choose to include this appendix as we believe could prove useful in other contexts or inspire future work that aim to further improve the quality of curvature approximations.

## J  CODE

Code is available at: https://github.com/Qualcomm-AI-research/llm-surgeon.