# OpenReview forum: "The LLM Surgeon"
_ICLR.cc/2024/Conference — ICLR 2024 poster_

### Official Review · Reviewer_2GDn · 2023-10-30

**Soundness:** 3 good
**Presentation:** 3 good
**Contribution:** 2 fair
**Rating:** 5
**Confidence:** 5

**Summary:**

This paper introduces LLM Surgeon, a method that enhances the scalability of Kronecker-factored curvature approximations of the targeted loss landscapes, designed for pruning LLMs at arbitrary sparse patterns. The authors demonstrate that this proposed methodology consistently improves the PPL performance of current methods across a range of LLMs.

**Strengths:**

1. The paper is well-organized and easy to comprehend.

2. The fundamental idea for estimating the curvature of the loss landscape is reasonable and innovative.

**Weaknesses:**

1. Since the authors adopt a multi-shot sparse approach, it would be beneficial to quantitatively compare the time costs and GPU memory consumption with SparseGPT.

2. While the authors emphasize sparsity in large models, the largest model they utilize is of 7-billion parameters. It might provide readers with a clearer view if experiments involving larger model sizes were included.

3. Global rank ordering is a sound strategy, but there seems to be a lack of an ablation experiment, that is, whether the suggested method outperforms SparseGPT under the same layer-wise sparsity situation.

4. Although the authors underlines that LLM surgeon can be migrated to structured pruning, and as the authors stated, "To the best of our knowledge, this is the first method to successfully perform structured pruning for LLMs," they do not discuss nor compare their approach to the already presented structured LLM pruning method[1]. This lack of discussion appears less meticulous. Furthermore, the authors should also consider comparing their work with another state-of-the-art large model sparse method, Wanda [2], which can also be adapted for pruning LLMs at any patterns.

[1] LLM-Pruner: On the Structural Pruning of Large Language Models. In NeurIPS, 2023
[2] A Simple and Effective Pruning Approach for Large Language Models. In Arxiv, 2023

**Questions:**

Please see the weakness part.

---

> ### Author Response · Authors · 2023-11-17
>
> Thank you for your feedback and help to improve the paper. In particular, we thank the reviewer for noting the paper is well-organised and easy to comprehend and deem our fundamental contributions reasonable and innovative.
>
> 1. compute and memory
>
> We agree with the reviewer that we should include a quantitative comparison between methods. Compared to other methods (e.g. SparseGPT), we are able to obtain better compression results but only at the expense of longer compression time. We will make sure this is absolutely clear from the main text. The table below shows compression time on Llama-v2 7B and 13B models. We will add these results to the paper appendices:
>
> |           | SparseGPT  (baseline)           | Unstructured LLM-Surgeon (ours) |
> | --------- | --------------------- | ------------------------ |
> | Llama-7B  | <5m / 1 GPU (32GB)    | 2d8h16m / 4xH100 80 GB   |
>
> |           | K-OBD   (baseline)              | Structured LLM-Surgeon (ours)   |
> | --------- | --------------------- | ------------------------ |
> | Llama-7B  | 16h58m / 4xH100 80 GB | 17h08m / 4xH100 80 GB    |
> | Llama-13B | 1d6h5m / 8xH100 80 GB | 1d9h26m / 8xH100 80 GB   |
>
> The method is highly parallelizable and more GPUs can be traded for a linear speed-up in time, if desired. Compression time measured using the required memory:
>
> |                                       |Runtime|PPL 90%|PPL 80%|PPL 70%|PPL 60%|PPL 50%|
> |---------------------------------------|-------|-------|-------|-------|-------|-------|
> |Unstructured baseline (SparseGPT)      |**<5m**    |5.49   |5.58   |5.71   |5.94   |6.51   |
> |Unstructured LLM Surgeon (ours)        |2d8h16m|**5.13**   |**5.20**   |**5.36**   |**5.66**   |**6.08**   |
> |Structured baseline (K-OBD)            |**16h58m** |5.48   |9.14   |15.43  |28.03  |46.64  |
> |Structured LLM Surgeon (ours)          |17h08m |**5.25**   |**6.18**   |**7.83**   |**10.39**  |**15.38**  |
> |Structured baseline (K-OBD) Llama-13B  |**1d6h5m** |4.908  |6.294  |10.08  |13.06  |16.06  |
> |Structured LLM Surgeon (ours) Llama-13B|1d9h26m|**4.692**  |**5.286**  |**6.207**  |**7.245**  |**9.428**  |
>
> Further engineering efforts may further speed up compression, especially for unstructured pruning. As mentioned, our method does come with a higher computational cost compared to other approaches. Importantly, compression of LLMs in general only needs to happen once after which a pruned model can be deployed infinitely many times without further cost. This motivates our method which takes longer to run but provides much better final test performance.
>
> 2. Larger models
>
> We agree with the reviewers that it would be interesting to assess performance on even larger models. The model sizes were chosen to demonstrate compression on OPT models with millions of parameters but also on the Llama architecture, demonstrating pruning of models in the order of billions of parameters. Following the suggestions by the reviewers (zzzW, 2GDn), we now also have results for the larger Llama-v2 13B model and evaluated its performance on downstream tasks (see results below). We find that our method also outperforms baselines on this larger model, as well as on downstream tasks.
>
> 3. Global rank ordering
>
> We thank the reviewer for raising this point. Our proposed improved curvature approximations use gradient information that relates curvatures of individual layers to the final global objective. The gradient terms can be interpreted as the relative influence layer outputs have on the final objective. For this reason, our improved curvature estimates are especially useful in conjunction with global thresholding, which allocates weight to be pruned globally rather than for layers separately. Nevertheless, we will run an ablation study on this and include it in the appendix.
>
> 4. Structured pruning related work
>
> We thank the reviewer for bringing other structured sparsification works to our attention. Both works are going to be published at NeurIPS 2023 and appeared a few months before the ICLR deadline on arXiv so should be considered concurrent work. Further, [2] does not seem to include any experiments on structured pruning of rows and columns and only considers semi-structured pruning (e.g. 2:4). The other paper [1] does include such experiments, but for -20% sparsity on Llama-v2 reports a very large degradation in performance from 12.62 to 74.63, whereas our method on the same model, sparsity and data achieves test perplexity from 5.12 to 6.18. We would like to emphasise that structured pruning is a very hard task on which our method outperforms any existing approach by a large margin. Regardless, we deem [1, 2] relevant prior literature and will add them to the related work section.
>
> [1] LLM-Pruner: On the Structural Pruning of Large Language Models. In NeurIPS, 2023
>
> [2] A Simple and Effective Pruning Approach for Large Language Models. In Arxiv, 2023
>
> ** Edit 20 Nov: added timing results for Llama-v2 13B and GPU memory

---

> > ### Author Response · Authors · 2023-11-20
> > **larger models [additional results]**
> >
> > 2. larger models [additional results]
> >
> > Following the suggestions by the reviewers (zzzW, 2GDn), we have also evaluated structured compression on a larger Llama-v2 13B model and evaluated its performance on downstream tasks. Due to the short time frame of the discussion period and resource constraints, we can now only show Llama-v2 13B results, but we intend to add results for even larger models to the camera ready version of the paper. Test perplexities (lower is better) can be found in table below:
> >
> > |         | Dense 100% | 90%   | 80%   | 70%   | 60%   | 50%   |
> > | ------- | ---------- | ----- | ----- | ----- | ----- | ----- |
> > | K-OBD   | 4.547      | 4.908 | 6.294 | 10.08 | 13.06 | 16.06 |
> > | LLM Surgeon | 4.547      | 4.692 | 5.286 | 6.207 | 7.245 | 9.428 |
> >
> > as well as evaluated results on downstream benchmarks (higher is better):
> >
> > | Llama-v2 13B      |     |wikitext (word ppl)|boolq |piqa  |hellaswag|winogrande|arc_easy|arc_challenge|openbookqa|copa  |lambada_openai|wsc273|AVERAGE|
> > |--------------|-----|-------------------|------|------|---------|----------|--------|-------------|----------|------|--------------|------|-------------|
> > |Dense baseline|100% |8.23               |80.52%|80.52%|79.38%   |72.14%    |77.53%  |49.23%       |45.20%    |90.00%|76.77%        |89.38%|74.07%       |
> > |              |     |                   |      |      |         |          |        |             |          |      |              |      |             |
> > |LLM Surgeon   |90%  |8.57               |81.07%|79.87%|79.24%   |72.38%    |76.30%  |49.91%       |47.20%    |92.00%|75.47%        |89.38%|74.28%       |
> > ||80%           |10.08|80.86%             |79.00%|77.09%|70.56%   |75.93%    |46.76%  |46.80%       |90.00%    |67.79%|86.45%        |72.12%|
> > ||70%           |12.74|74.50%             |76.50%|71.52%|68.67%   |69.74%    |40.27%  |45.00%       |91.00%    |54.40%|83.52%        |67.51%|
> > ||60%           |16.00|64.62%             |73.01%|65.04%|65.75%   |63.80%    |37.12%  |39.60%       |90.00%    |44.50%|81.32%        |62.48%|
> > ||50%           |23.75|65.66%             |68.77%|56.19%|63.22%   |56.19%    |31.83%  |36.60%       |85.00%    |35.16%|77.29%        |57.59%|
> > |              |     |                   |      |      |         |          |        |             |          |      |              |      |             |
> > |K-OBD         |90%  |8.79               |81.31%|79.76%|79.12%   |72.22%    |76.94%  |47.95%       |47.80%    |91.00%|75.26%        |88.64%|74.00%       |
> > ||80%           |11.79|80.80%             |79.16%|76.80%|70.56%   |73.74%    |46.93%  |48.60%       |88.00%    |58.99%|87.55%        |71.11%|
> > ||70%           |20.00|66.76%             |74.43%|64.18%|64.96%   |56.23%    |36.01%  |39.00%       |88.00%    |38.54%|79.49%        |60.76%|
> > ||60%           |27.74|55.66%             |70.24%|55.52%|60.46%   |49.62%    |32.68%  |35.80%       |80.00%    |30.06%|73.63%        |54.37%|
> > ||50%           |37.38|59.79%             |66.54%|48.39%|57.46%   |46.59%    |30.72%  |34.00%       |77.00%    |24.61%|69.96%        |51.50%|
> >
> > We observe that our model also outperforms baselines on existing Llama-v2 13B models. We stress that these results are obtained on structured pruning of rows and columns, which can be regarded as the hardest and most constrained pruning structure. Yet, our proposed LLM surgeon is able to compress Llama 13B by 20% with less than 2% drop in downstream task performance in this challenging setup. It also significantly outperforms the baseline for all compression rates, both in terms of test perplexity and downstream task performance.

---

> > > ### Comment · Reviewer_2GDn · 2023-11-21
> > > **Thanks for the response**
> > >
> > > Thanks for the author's responses and efforts, which partially addressed my concerns. However, my unease about the efficiency of the proposed method persists. It is excessively costly for a LLM pruning method to compress llama-7b requiring 4xH100 80 GB and 2d8h16m, respectively. For instance, under the same conditions, SparseGPT only takes minutes, and Wanda even takes mere seconds with single GPU. While I comprehend the authors' assertion that their method, though more time-consuming, yields better performance, I question why we cannot directly use lora fine-tuning for performance recovery? As reported in [1], a 50% sparse llama7b can be elevated from a PPL of 7.26 to 6.84 merely by 4h of fine-tuning. This is akin to the performance gap between llm-surgen and SparseGPT, not to mention that lora only requires one GPU. Consequently, although I acknowledge the authors' comprehensive response, my apprehension concerning the method's efficiency prevents me from recommending this paper for acceptance.
> > >
> > > [1] A Simple and Effective Pruning Approach for Large Language Models. In Arxiv, 2023

---

> ### Author Response · Authors · 2023-11-21
>
> We are very glad that most of the concerns have been addressed. In this reply, we hope to take away the remaining hesitation regarding model efficiency.
>
> Overall, our method was developed for structured pruning, where it shines. Here we are roughly as fast as the baseline (K-OBD) and much better (17h08m versus 16h58m on Llama-v2). For unstructured, it is also better but indeed a bit slow, but this is not the main point of the method. We included this comparison to have a well-established baseline to compare to and because the method outperformed pruning performance on all structure types.
>
> - why not replace costly pruning with finetuning
>
> This is indeed an important question. However, we would like to push back on the possibility of using finetuning as an alternative for our approach. It is indeed that for unstructured and semi-structured pruning finetuning can be beneficial and can sometimes work [1], although not always reliably. The reason for this is that limited data used for compression is not always effective in preserving the capabilities of an existing pre-trained model. Empirically, we found that first-order updates (e.g. LoRA) can not always make up for removals of weights (see also Sec. 4.2), a problem which gets much worse in the case of structured pruning, in which the degradation in loss can be very large if weights are removed without updates that take weight correlations into account.
>
> Compared to finetuning, our approach has two big benefits, namely: (a) it is hyperparameter free, (b) allows dynamic allocation which weights to be removed, and (c) correlated weight updates which allow retaining as much information as possible from the existing pre-trained model, even with limited data.  In experiments, we found that finetuning to not sufficient to recover performance and are not aware of literature using finetuning to work for structured pruning. We believe this difficulty to be the reason why relatively few papers even consider structured pruning in their experiments. As such, our method has important benefits compared to finetuning approaches. We do think we could have made this point more clear in the paper and will improve this.
>
> - computational cost
>
> Our method takes longer to run but achieves higher performance.
>
> We very much admire efforts such as [1] or SparseGPT as these works demonstrate how much compression can be achieved only using activations and weights. Nevertheless, only relying on activations and weight strength implies strong assumptions on the curvature of the loss and makes these methods unsuitable for more constrained structure types. Our improved curvature allows for dynamic allocation of weight removal and improved correlated weight updates. As a result, our method can be used for structured pruning and results in higher overall performance.
>
> Although we are proud of the efficiencies that allow scaling very rich Kronecker structures to LLMs, our resulting method does remain much more computationally expensive compared to existing pruning methods in relative terms. From a practical perspective, however, we argue that the compute time may not be 'excessive' at all! The reason for this is that a model only needs to be pruned once after which it can be deployed infinitely many times. Therefore, it can very well be economically advantageous to temporarily spend multiple GPU resources on compression to achieve a better performance after pruning. In absolute terms, the use of multiple large GPUs is very common practice in the field of large language models and many more GPUs are typically used to train and deploy large language models. Moreover, our method can be trivially parallelized in case further speed-ups or larger models are required. We hope this provides a better context and emphasises the trade-off that should be made by the practitioner.
>
> Further, the trade-off between performance and computational cost might very well be fundamental (therefore something that we may not be able to fix). Given that, we sincerely hope that the computational cost is not a ground for rejection, as our method offers significant benefits including structured pruning and state-of-the-art pruning performance.
>
> We thank the reviewer and do agree that the trade-off should be communicated clearly to the reader and practitioners who may wish to use the method. We promise to make an effort to ensure this is very clear from the main text.

---

### Official Review · Reviewer_zzzW · 2023-11-01

**Soundness:** 3 good
**Presentation:** 3 good
**Contribution:** 3 good
**Rating:** 5
**Confidence:** 4

**Summary:**

The paper proposes a method called "LLM Surgeon" for efficient pruning and compression of large pretrained language models like OPT and LLAMa. It scales up second-order Hessian-based pruning methods like Optimal Brain Surgeon using Kronecker-factored approximations of the Fisher information matrix. It derives closed-form solutions for removal costs and correlated weight updates when pruning multiple weights jointly. Experiments show the method can prune OPT and LLAMa models by 20-30% with minor performance loss and outperforms prior work.

**Strengths:**

- The paper is well-written and clearly presented;
- The paper provides a general pruning framework applicable to different structured and unstructured schemes.
- The paper provides careful derivation of update rules that consider correlations between weights, theoretically principled and extends classical segundo-order pruning methods.
- The proposed methods achieves state-of-the-art pruning results on large language models, especially for structured pruning.
- Detailed ablation regarding the low-rank components, approximation methods, as well as the qualitative sparsity level analyses are provided to show the comprehensiveness of the proposed methods and design choices;

**Weaknesses:**

- The paper still uses approximations for computational tractability which limits pruning performance.
- Structured pruning leads to irregular sparsity patterns which are difficult to accelerate. The real inference speedup or memory savings after pruning is unknown;
- Additional FLOPs for approximation and updating offsets gains during deployment are needed, while the detailed comparison and discussion might be missing.
- Some related works might also be good to include [1];

[1] Yu, Shixing, et al. "Hessian-aware pruning and optimal neural implant." Proceedings of the IEEE/CVF Winter Conference on Applications of Computer Vision. 2022.

**Questions:**

- Could the authors provide both inference speed and additional cost for approximation and updating the offsets?
- Could larger model sizes also be included and evaluated throughout?
- Besides the PPL, could the author also provide the 0shot performance degradation regarding the OPT/LLaMA models for a comprehensive evaluation;

---

> ### Author Response · Authors · 2023-11-17
>
> Thank you for your feedback and help to improve the paper. We thank the reviewer for finding the paper is well written, clearly presented and theoretically principled. We further value that the reviewer appreciated the 'careful derivations of correlated update rules' and noted the strength of the work and state-of-the-art pruning results, especially on structured pruning.
>
> > Use of approximations
>
> Due to the high dimensionality of LLM models, the constraint optimization problem considered in this work is inherently intractable. This means that it can be mathematically proven that the problem can not be solved in an exact manner and the sheer use of an approximation should thus not be considered a 'weakness' but rather a necessity. In fact, most other methods (even magnitude pruning) can be seen as some sort of crude approximation to the Hessian of the global loss and are strictly more approximate than ours. Thus, compared to other methods we use the most accurate curvature approximation. We expect this to be a misunderstanding and will make this more clear in the main text to prevent confusion.
>
> > Irregular sparsity patterns
>
> We thank the reviewer for raising this as the computational benefits of the considered pruning structures might not have been emphasised enough in the paper. It is not true that structured pruning results in irregular sparsity patterns. Rather the opposite, it provides the most regular sparsity patterns - hence the name 'structured' - and is very easy to accelerate. As complete rows and columns are removed, structured pruning of X% of weights means that matrix dimensions can directly be reduced by X%. The resulting smaller matrices have simply lower dimensions and are thus directly amenable to existing standard CUDA/and hardware-accelerated kernels. We recognize that we only very briefly mention this in Sec. 2. We thank the reviewer and will make sure this point is made more clear in the main text.
>
> > Savings in terms of FLOPS / compute time
>
> We are not exactly sure what is meant by 'updating offsets gains', but will assume the question is whether reported weight reduction relates to actual computational and memory savings at deployment. Unlike most other sparsity patterns, structured pruning allows for a direct reduction in the dimensionality of weight matrices by removing complete rows and columns. Therefore, a reduction of weights will result in the same reduction in the number of FLOPS. This is not generally true for most other sparsity patterns, and in fact an important benefit and in part the motivation for structured sparsity patterns. Below, we provide compression time of compressed models which we will add to the paper appendices.
>
> > Inference time overhead
>
> A major advantage of our method is that there is no overhead at inference time. All savings are absorbed into the weight of the neural network and there are no separate offsets to be separately stored. We have not mentioned this benefit explicitly and will make sure it is clear in the paper.
>
> > Related work
>
> We thank the reviewers for pointing out this work. We find the reference relevant and will add it to an updated related work section.
>
> > Questions / additional experiments
>
> 1. Performance
>
> The table below shows compression runtimes on the Llama-7b model. We will add these results to the paper appendices.
>
> | |Runtime|PPL 90%|PPL 80%|PPL 70%|PPL 60%|PPL 50%|
> |-|-|-|-|-|-|-|
> |Unstructured baseline (SparseGPT) Llama-v2 7B |**<5m**|5.49|5.58|5.71|5.94|6.51|
> |Unstructured LLM Surgeon (ours)  Llama-v2 7B  |2d8h16m|**5.13**|**5.20**|**5.36**|**5.66**|**6.08**|
> |Structured baseline (K-OBD)  Llama-v2 7B  |**16h58m** |5.48|9.14|15.43|28.03|46.64|
> |Structured LLM Surgeon (ours)  Llama-v2 7B |17h08m |**5.25**|**6.18**   |**7.83**   |**10.39**|**15.38**|
> |Structured baseline (K-OBD) Llama-v2 13B|**1d6h5m** |4.908|6.294  |10.08  |13.06  |16.06|
> |Structured LLM Surgeon (ours) Llama-v2 13B|1d9h26m|**4.692**|**5.286**|**6.207**|**7.245**|**9.428**|
>
> Our method is most efficient for structured pruning, but it must be noted that engineering efforts may further improve speed for unstructured pruning. The focus of the paper is structured pruning, on which we achieve state-of-the-art compression rates. Importantly, compression of LLMs only needs to happen once after which a pruned model can be deployed infinitely many times without further cost. This motivates our method which takes longer to run but provides much better final test performance. We will add the timing results to the paper and make the last point clear from the text.
>
> 2. Large model sizes
>
> We agree that it would be interesting to assess performance on even larger models. Following the suggestions by the reviewers (zzzW, 2GDn), we now also include results for the larger Llama-v2 13B model and evaluated performance on downstream tasks (see reply to Reviewer 2GDn). We find that our method also outperforms baselines on this larger model, as well as on downstream tasks.

---

> > ### Author Response · Authors · 2023-11-17
> >
> > 3. 0shot performance
> >
> > As also mentioned by other reviewers, we recognise the need for extended evaluation beyond perplexity scores. Therefore, we have performed additional downstream benchmarks on structured pruning task and show results in the table below:
> >
> > | Structured pruning (compressed with wikitext-2) |      | wikitext word ppl | boolq | piqa  | hallaswag | winogrande | arc_easy | arc_challenge | openbookq | copa  | lambada_openai | wsc273 | AVERAGE wikitext2 |
> > | -------------------------------------- | ---- | ----------------- | ----- | ----- | --------- | ---------- | -------- | ------------- | --------- | ----- | -------------- | ------ | ----------------- |
> > | Dense baseline                         | 100% | 9.24              | 77.74 | 79.11 | 75.99     | 69.14      | 74.58    | 46.25         | 44.20     | 86.00 | 73.92          | 85.71  | 71.26             |
> > |                                        |      |                   |       |       |           |            |          |               |           |       |                |        |                   |
> > | LLM Surgeon                            | 90%  | 9.63              | 76.21 | 78.56 | 75.39     | 67.64      | 74.12    | 46.50         | 43.60     | 85.00 | 72.64          | 84.98  | 70.46             |
> > |                                        | 80%  | 12.16             | 72.97 | 77.09 | 71.30     | 66.30      | 71.36    | 41.89         | 41.80     | 87.00 | 56.43          | 80.22  | 66.66             |
> > |                                        | 70%  | 16.91             | 61.25 | 73.56 | 60.72     | 61.09      | 63.09    | 36.69         | 38.80     | 81.00 | 28.33          | 76.56  | 58.11             |
> > |                                        | 60%  | 25.15             | 44.98 | 69.26 | 48.04     | 54.38      | 52.31    | 30.29         | 36.80     | 78.00 | 11.72          | 68.50  | 49.43             |
> > |                                        | 50%  | 43.68             | 39.60 | 64.36 | 40.29     | 52.57      | 44.91    | 26.28         | 30.80     | 74.00 | 6.52           | 61.54  | 44.09             |
> > |                                        |      |                   |       |       |           |            |          |               |           |       |                |        |                   |
> > | K-OBD                                  | 90%  | 9.89              | 76.67 | 78.02 | 74.80     | 68.11      | 75.17    | 46.33         | 44.60     | 86.00 | 72.71          | 82.78  | 70.52             |
> > |                                        | 80%  | 17.62             | 74.34 | 75.24 | 67.85     | 64.64      | 63.80    | 40.27         | 41.60     | 83.00 | 30.23          | 82.42  | 62.34             |
> > |                                        | 70%  | 32.72             | 65.29 | 71.82 | 53.07     | 56.83      | 51.05    | 33.11         | 37.80     | 79.00 | 12.21          | 70.70  | 53.09             |
> > |                                        | 60%  | 68.63             | 60.80 | 65.67 | 43.99     | 53.20      | 41.79    | 28.50         | 34.00     | 75.00 | 7.04           | 60.44  | 47.04             |
> > |                                        | 50%  | 136.33            | 61.56 | 60.66 | 36.84     | 53.04      | 36.11    | 26.71         | 33.00     | 72.00 | 4.70           | 61.17  | 44.58             |
> > |                                        |      |                   |       |       |           |            |          |               |           |       |                |        |                   |

---

> > > ### Author Response · Authors · 2023-11-17
> > >
> > > | Structured pruning (compressed with C4) |      | Wikitext word ppl | boolq | piqa  | hallaswag | winogrande | arc_easy | arc_challenge | openbookq | copa  | lambada_openai | wsc273 | AVERAGE wikitext2 |
> > > | ------------------------------- | ---- | ----------------- | ----- | ----- | --------- | ---------- | -------- | ------------- | --------- | ----- | -------------- | ------ | ----------------- |
> > > | Dense baseline                  | 100% | 9.24              | 77.74 | 79.11 | 75.99     | 69.14      | 74.58    | 46.25         | 44.20     | 86.00 | 73.92          | 85.71  | 71.26             |
> > > |                                 |      |                   |       |       |           |            |          |               |           |       |                |        |                   |
> > > | LLM Surgeon                     | 90%  | 9.90              | 77.03 | 78.45 | 74.95     | 68.27      | 73.19    | 45.99         | 44.60     | 84.00 | 72.81          | 82.78  | 70.21             |
> > > |                                 | 80%  | 14.42             | 75.60 | 76.82 | 69.71     | 63.85      | 70.29    | 41.30         | 42.80     | 87.00 | 45.53          | 82.42  | 65.53             |
> > > |                                 | 70%  | 25.16             | 66.39 | 72.85 | 58.11     | 56.83      | 62.16    | 34.47         | 38.40     | 80.00 | 22.69          | 69.96  | 56.19             |
> > > |                                 | 60%  | 45.35             | 62.48 | 68.93 | 48.10     | 55.64      | 51.56    | 27.99         | 35.20     | 70.00 | 12.56          | 61.54  | 49.40             |
> > > |                                 | 50%  | 77.30             | 62.60 | 65.02 | 41.70     | 54.22      | 42.55    | 24.23         | 31.20     | 71.00 | 7.26           | 60.44  | 46.02             |
> > > |                                 |      |                   |       |       |           |            |          |               |           |       |                |        |                   |
> > > | K-OBD                           | 90%  | 10.59             | 75.47 | 78.18 | 73.61     | 66.46      | 72.52    | 44.37         | 43.60     | 87.00 | 71.22          | 82.42  | 69.48             |
> > > |                                 | 80%  | 20.12             | 73.36 | 75.14 | 66.11     | 62.43      | 62.84    | 38.23         | 41.00     | 86.00 | 21.50          | 78.39  | 60.50             |
> > > |                                 | 70%  | 56.92             | 63.30 | 68.44 | 52.31     | 55.64      | 46.72    | 31.31         | 34.60     | 77.00 | 5.69           | 68.86  | 50.39             |
> > > |                                 | 60%  | 112.85            | 62.23 | 64.47 | 46.36     | 52.17      | 40.53    | 29.52         | 32.40     | 72.00 | 2.91           | 63.00  | 46.56             |
> > > |                                 | 50%  | 272.16            | 62.42 | 61.70 | 38.47     | 50.43      | 33.29    | 26.96         | 31.80     | 65.00 | 0.91           | 59.34  | 43.03             |

---

### Official Review · Reviewer_zF62 · 2023-11-02

**Soundness:** 3 good
**Presentation:** 3 good
**Contribution:** 3 good
**Rating:** 5
**Confidence:** 3

**Summary:**

This paper proposes a way to prune the weights of a pretrained LLM with a negligible loss in performance by iteratively solving a quadratic optimization problem using the curvature of a local minimum. To save the memory of materializing hessian, the work calculates the covariance layer-wise with Kronecker factorization. Also, the curvature is calculated incrementally and the remaining weights are corrected with a first-order term as more weights are pruned so that the weight remains in a local minimum.

**Strengths:**

The approach is sound and is presented well. Not having to materialize the hessian makes this amenable for LLM pruning.

**Weaknesses:**

While it is believable that this method would generalize, the setup is unsatisfying in that the dataset used for compression is drawn from wikitext-2 which is a narrow domain (meaning the loss landscape may be easier to optimize and prune than a broader distribution), and the final model is evaluated only on the test perplexity of the same dataset. SparseGPT uses C4 and reports downstream performance on various standard benchmarks. The paper could really strengthen its claim by repeating the same setup as SparseGPT. Downstream benchmarks are required for a fair comparison and acceptance of the work.

**Questions:**

Equation 2 should read `- log(D | theta)`. There is a typo in `General solution  We denote ... e_{q_k}` => `e_{k_q}`.

Related Work could include more previous works on LLM compression.

---

> ### Author Response · Authors · 2023-11-17
>
> Thank you for your feedback and help to improve the paper. We appreciate that the reviewer found the approach sound and well presented, further recognising our contributions to Hessian-based pruning amenable to LLMs.
>
> > Training with C4 instead of wikitext2
>
> Unlike SparseGPT which used the C4 dataset, we used the wikitext2 dataset for compression in our experiments. Although the amount of data is the same, C4 can be regarded slightly more general in that it contains multiple languages. We agree with the reviewer that it would be interesting to evaluate performance with C4. Following these comments, we have therefore also evaluated the use of the C4 dataset as reference data for compression in the case of structured pruning. We find that our model in this case still outperforms our baselines.
>
> **The additional results on downstream tasks can be found in our reply to Reviewer zzzW (also includes training with suggested C4 data)**
>
> Further, we would like to stress that all results in our paper (including the SparseGPT) were trained on wikitext2 to enable fair comparison. To do so, we used the same dataloader and evaluation script as the official SparseGPT repo and reran all SparseGPT results to be trained on wikitext2. This even resulted in better scores for our SparseGPT baseline than the C4 trained results from the original SparseGPT paper. Yet, we find that our method using improved curvature estimates still outperformed the baselines. We are aware that these efforts might not have been made clear enough and will ensure this is clearly stated in the final version.
>
> > Downstream benchmarks
>
> The reviewer raised the need for downstream benchmarks 'for a fair comparison and acceptance of the work'. Although perplexity scores in prior work tend to correlate well with downstream performance, we recognise the need for such additional experiments to offer a complete comparison between methods. We have performed downstream benchmarks on structured pruning compressed using wikitext-2 but also the C4 dataset.
>
> **The additional results on downstream tasks can be found in our reply to Reviewer zzzW**
>
> We find that our method not only performs well in terms of test perplexity but also correlates well with downstream performance, outperforming the baselines on these downstream tasks.
>
> > Related work
>
> We agree with the suggestion by the reviewer to extend the related work so that all important pruning work on large language model compression is included. In particular, we will include related work on structured pruning and other LLM pruning works that use a curvature-based approach.
>
> This is indeed a typo. Typos will be fixed.

---

### Author Response · Authors · 2023-11-20
**General response**

Dear Reviewers and ACs,

We would like to thank the reviewers for the constructive feedback. All reviewers found the paper well-written, sound and recognised our theoretically principled contributions that led to the improved general method for pruning large language model, outperforming existing approaches. We would emphasise that our method even achieves state-of-the-art on structured pruning of rows and columns, which is regarded as the hardest and most constrained pruning structure.

In general, we agreed with the feedback given by reviewers. We hope to have addressed all existing concerns and have provided the additional results required for acceptance. In particular:

- evaluating downstream task performance (reviewers zF62, zzzW)
- ablation of compressing using C4 versus wikitext-2 (reviewer zF62)
- results on larger model using the Llama-v2 13B architecture (reviewers zzzW, 2GDn)
- providing computational and memory requirements (reviewers zzzW, 2GDn)

We thank the reviewers for their help and feedback to improve the paper. We would appreciate a response to the rebuttal in case of any follow-up questions or concerns you might have.

Sincerely,
Authors

---

### Meta-Review · Area_Chair_Heh7 · 2023-12-06

**Metareview:**

Reviewer zzzW summarized it well:

> The paper proposes a method called "LLM Surgeon" for efficient pruning and compression of large pretrained language models like OPT and LLAMa. It scales up second-order Hessian-based pruning methods like Optimal Brain Surgeon using Kronecker-factored approximations of the Fisher information matrix. It derives closed-form solutions for removal costs and correlated weight updates when pruning multiple weights jointly. Experiments show the method can prune OPT and LLAMa models by 20-30% with minor performance loss and outperforms prior work.

This is a nice paper that (in some sense) generalizes methods such as SparseGPT using efficient approximations to second order information. The empirical gains over existing baselines are clear, and given that this is the case in structured pruning as well, this approach can enable actual speed-ups. On the negative side, the proposed approach is substantially slower than baselines, and moreover, experiments are not performed on the largest (e.g., 70B) models. But given the methodological contribution combined with (some) empirical gains, I think this paper should be accepted.

**Justification For Why Not Higher Score:**

The method is substantially slower than simpler baselines.

**Justification For Why Not Lower Score:**

The mathematical and methodological contribution of the work is substantial. Moreover, while the proposed approach is slower than baselines (SparseGPT/Wanda), the outperformance over these baselines at the same sparsity rates is significant.

---

### Decision · Program_Chairs · 2024-01-16

Accept (poster)